# LRIM: A Physics-Based Benchmark for Provably Evaluating Long-Range Capabilities in Graph Learning

**Joël Mathys**
ETH Zurich
`jmathys@ethz.ch`

**Henrik Christiansen**[1]**, Federico Errica**[1]**, Takashi Maruyama**[2] **& Francesco Alesiani**[1]
NEC Laboratories Europe
[1]`{firstname.lastname}@neclab.eu`, [2]`49takashi@nec.com`

## Abstract

Accurately modeling long-range dependencies in graph-structured data is critical for many real-world applications. However, incorporating long-range interactions beyond the nodes' immediate neighborhood in a *scalable* manner remains an open challenge for graph machine learning models. Existing benchmarks for evaluating long-range capabilities either cannot *guarantee* that their tasks actually depend on long-range information or are rather limited. Therefore, claims of long-range modeling improvements based on said performance remain questionable. We introduce the Long-Range Ising Model Graph Benchmark, a physics-based benchmark utilizing the well-studied Ising model whose ground truth *provably* depends on long-range dependencies. Our benchmark consists of ten datasets that scale from 256 to 65k nodes per graph, and provide controllable long-range dependencies through tunable parameters, allowing precise control over the hardness and "long-rangedness". We provide model-agnostic evidence that local information is insufficient, further validating the design choices of our benchmark. Via experiments on classical message-passing architectures and graph transformers, we show that both perform far from the optimum, especially those with scalable complexity. Our goal is that our benchmark will foster the development of scalable methodologies that effectively model long-range interactions in graphs.

## 1 Introduction

Since the early days of deep learning on graphs (Sperduti & Starita, 1997; Micheli & Sestito, 2005; Gori et al., 2005; Scarselli et al., 2009; Micheli, 2009), researchers have tried to automatically learn a task-defined mapping given graph-structured data. At the very heart of the most popular architecture, namely Graph Neural Networks (GNNs), is the idea that repeated aggregation of local information expands the "receptive field" of each node (Bacciu et al., 2020) in a way similar to convolutional neural networks for images (LeCun et al., 1995). Such an expansion is crucial, for instance, in tasks where the true mapping requires a non-local processing of information among nodes in the graph. In this case, researchers typically talk about *capturing long-range dependencies*.

Long-range dependencies or interactions are an important component of physics and chemistry, manifesting, for example, in quantum systems (Defenu et al., 2023), protein folding (Gromiha & Selvaraj, 1999) or astronomy (Carroll & Ostlie, 2017). An example from biology is mRNA splicing, a fundamental part of the gene expression process: splicing is inhibited if we "disable" long-range dependencies between distant regions of mRNA (Rüegsegger et al., 2001). The protein binding mechanism, whose understanding is crucial for the development of vaccines, also depends on long-range interactions between proteins (Ferber et al., 2012).

As of today, most deep learning models on graphs struggle to incorporate long-range interactions. For GNNs, the reason is tied to their efficient but limited message-passing scheme: expanding the

receptive field to capture long-range information rapidly increases the amount of information that every node has to process and store, generating a *computational bottleneck* (Arnaiz-Rodriguez & Errica, 2025). When addressing the long-range limitations of existing models, the first step should be to benchmark novel methods on tasks that provably depend on long-range information. Unfortunately, while popular existing benchmarks focus on real-world data (Dwivedi et al., 2022), they cannot **guarantee** that the task to solve hinges on long-range dependencies (Tönshoff et al., 2023). The reason is that the definition of the task is *unknown*, which is almost always the case in machine learning.

We introduce the Long-Range Ising Model (LRIM) Graph Benchmark [1] a *provable* and controllable long-range benchmark based on the well-studied and fundamental Ising model for graph learning. The Ising model was originally introduced in statistical physics to study magnetic materials (Peierls, 1936) and fundamental properties related to phase transitions (Wilson, 1983; Stanley, 1987; Yeomans, 1992). Over the decades, the influence of the model has spread to aid understanding and analyzing complex phenomena far beyond the original intention, with applications in protein folding (Bryngelson & Wolynes, 1987), percolation (Balogh et al., 2012), the theory of disordered systems (Parisi, 2023), or social systems such as stock markets (Durlauf, 1999) to name only a few. The tunable parameters of the LRIM control the dependency of each spin's energy on distant spins, thereby allowing us to control the impact of long-range dependencies; in other words, through the underlying physics *we can easily control the "long-rangedness" of the task*.

In the following, we introduce ten novel LRIM datasets scaling from 256 to 65k nodes, each providing provable and controllable long-range dependencies through the underlying physical model. Besides the inherent long-range dependency in the task formulation, we ensure that we sample configurations having large spatial correlations between node features by simulating at the pseudo-critical temperature. In addition, we demonstrate the validity of our approach by providing baseline-agnostic evidence. We analyze the behavior of oracle predictors restricted to local neighborhoods, which systematically degrade performance. Moreover, this consistent behavior provides a valuable continuous feedback signal for approaches, both for performance evaluation and during training. The analysis of our proposed benchmark through the lens of recently proposed long-range metrics (Bamberger et al., 2025) further supports our claims. Finally, we also empirically quantify the performance of message-passing architectures and graph transformers on our benchmark, revealing substantial gaps, especially when factoring in the computational cost of the methods. As such, our aim is that the LRIM Graph Benchmark provides the graph learning community with a powerful tool for evaluating and advancing long-range capabilities in a provable, controllable manner.

## 2 RELATED WORK

The literature on long-range benchmarks for graph learning can be roughly divided into two main directions. The first focuses on real-world datasets in which long-range interactions should occur, often related to phenomena in biology or chemistry. The most popular one is perhaps the Long-Range Graph Benchmark (LRGB) by Dwivedi et al. (2022), which consists of image segmentation tasks – adapted to graphs – together with peptides' function classification and property regression tasks. While this work identifies properties that should be required for long-range candidates, such as sufficiently large graphs, there appears to be no conclusive evidence that the proposed tasks require LR interactions. Even if they did, it remains unclear how to assess their impact and necessity for the empirical performance on these datasets. The benchmark served as a common test-bed to compare graph transformers (Müller et al., 2024), which seemed to achieve superior performance compared to classical GNNs. However, such empirical claims were later revised (Tönshoff et al., 2023; Bechler-Speicher et al., 2025), showing that with a proper hyper-parameter tuning, both techniques achieve similar performance on the peptide tasks. As such, LRGB still provides a valuable test-bed on real-world data, but any claim about long-range capabilities cannot be guaranteed.

On the other hand, there exist a variety of synthetic datasets to evaluate long-range capabilities of machine learning models. These often focus on gathering or copying information (Gu et al., 2020; Di Giovanni et al., 2023) beyond a certain distance or predicting non-local graph properties such as connectivity, distances or eccentricity (Corso et al., 2020; Rampášek & Wolf, 2021; Liang et al., 2025). Similarly, the GLoRa benchmark (Zhou et al., 2025) provides a detailed analysis of these

---

[1]Project is available at https://github.com/iJorl/lrim_graph_benchmark

datasets and proposes a new task that provably depends on specific lengths. While this provides a promising direction towards a more principled evaluation, these datasets still exhibit significant limitations: most prediction tasks exhibit a binary-like feedback signal. This results in models either perfectly solving the task or achieving null performance. Contrarily to these works, we propose a prediction task that: *i)* is central to the simulation of physical systems, as the Ising model has historically played a crucial role in advancing research and understanding of many fields (Budrikis, 2024); *ii)* provides evidence for the required long-range interaction; *iii)* allows for a more fine-grained and controllable continuous feedback signal both during training and evaluation.

Lastly, we mention that heterophilic datasets have often been believed to require long-range capabilities of GNNs. However, this viewpoint was recently criticized in Arnaiz-Rodriguez & Errica (2025) by providing clear counterexamples that the task might induce heterophily regardless of the nature of the problem.

## 3 BACKGROUND

The description of systems with many interacting components is ubiquitous in the natural and social sciences. For example, in nature, a colony of ants evolves based on interactions among the individual ants and based on their genetic inclinations, or in a transportation network, vehicles move, combining individual objectives and traffic rules. Using domain-specific modeling approaches, these systems can be investigated on a case-by-case basis using convoluted system prototypes that are often difficult to understand and interpret. Conversely, spin models, such as the Ising model, have proven powerful in describing relevant features of real systems while retaining simplicity. Based on simple microscopic interaction laws, they show rich emergent behavior, non-trivial phase-transitions and spin-spin correlations. Formally, the LRIM is defined by the graph topology $G$ (excluding self-interactions) and the Hamiltonian

$$\mathcal{H}(\{s_i\}) = -\frac{1}{2} \sum_{ij \in G} J_{ij} s_i s_j, \text{ with power-law potential } J_{ij} = \frac{1}{r_{ij}^{d+\sigma}}, \tag{1}$$

where $r_{ij} = |\mathbf{r}_j - \mathbf{r}_i|$ is the distance between two spins $s_i = \pm 1$ located at coordinate $\mathbf{r}_i$. The exponent $\sigma$ controls how long-ranged the interactions are. In our case, we consider the spins to be located on a regular $d = 2$ dimensional lattice. In general, the choice of $J_{ij}$ determines the model class: If instead one chooses nearest-neighbor interactions, one recovers the standard short-range lattice models ($J_{ij} = 1$ and $G = \{ij | r_{ij} = 1\}$), and random $J_{ij}$ define spin glasses such as the Sherrington–Kirkpatrick model (Sherrington & Kirkpatrick, 1975). We consider the system to be in contact with its thermal environment, that is, a canonical setting in which microscopic configurations follow the Boltzmann distribution

$$P(\{s_i\}) = \frac{1}{\mathcal{Z}} \exp\left(-\frac{\mathcal{H}(\{s_i\})}{k_b T}\right), \tag{2}$$

where the Boltzmann constant is set to unity ($k_b = 1$), $T$ is the temperature, and $\mathcal{Z}$ the (unknown) normalization constant.

This model has three distinct phases as a function of $T$, distinguished by the critical temperature $T_c$: (i) for $T > T_c$ the system is *disordered*, (ii) for $T = T_c$ it is *critical*, and (iii) for $T < T_c$ it is *ordered*. For finite systems as considered in our benchmark, the relevant temperature is not $T_c$ directly, but rather the *pseudo-critical temperature* characterizing the transition for the given system size $L$ (see Appendix B). It is known that at $T_c$ the correlations in the system diverge, i.e., spins are correlated across all distances, and no length scale dominates. As a consequence, *self-similar* structures emerge, and the spin clusters become *fractal*. More formally, in our setting, the connected correlation function is defined as

$$C(\mathbf{r}) = \langle s_i s_j \rangle - \langle s_i \rangle \langle s_j \rangle \sim \frac{1}{r^\eta} \text{ for } T = T_c \text{ and } r \to \infty,$$

where $\langle . \rangle = \mathbb{E}_{P(\{s_i\})}[.]$ symbolized expectations under the Boltzmann distribution of Eq. 2 and includes an average over the system. At $T_c$ and in the limit of large distances $r$ one has that $C(\mathbf{r})$ decays algebraically with the universal critical exponent $\eta$ and the corresponding correlation length diverges. This implies that **spins are correlated over large distances in a non-trivial way and,**

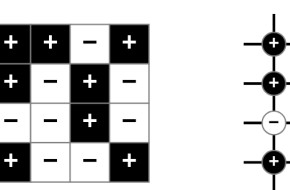 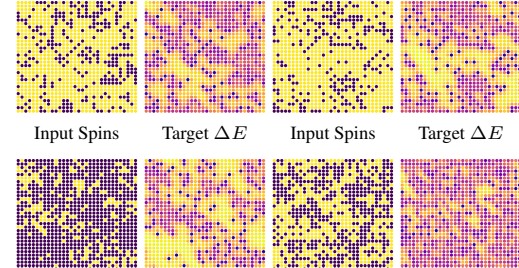

Figure 1: A $4 \times 4$ Ising spin configuration (left) is represented as an attributed graph (middle) in the LRIM Graph Benchmark. (Right) Four representative 32x32 spin configurations from the LRIM-32 dataset and their corresponding $\Delta E$ energy landscapes, demonstrating the non-trivial energetic landscape which provably depends on long-range interactions.

**in addition, are interacting via a long-range potential by construction**, posing an ideal way to construct appropriate configurations for our benchmark.

The value of $\sigma$ in Equation (1) controls how long-range the spins interact, and as a consequence, changes the equilibrium behavior of the model. For $\sigma < 1$ one has mean-field behavior and hence $\eta = 1$ (Kadanoff, 2009), for $1 < \sigma < \sigma_\times$, the critical exponents $\eta$ depends on $\sigma$ as $\eta = 2 - \sigma$, and for $\sigma_\times < \sigma$, one is in the short-range (nearest-neighbor) universality class with $\eta = 1/4$. The value of $\sigma_\times$ is discussed in the literature (Picco, 2012; Angelini et al., 2014; Shiratani & Todo, 2024; Liu et al., 2025), with $\sigma_\times = 2$ (Fisher et al., 1972) or $\sigma_\times = 1.75$ (Sak, 1973) being the most likely candidates. To simulate the system, a Markov Chain Monte Carlo simulation can be set up by proposing a random single spin flip $\{\ldots, s_i, \ldots\} \to \{\ldots, -s_i, \ldots\}$, and accepting the new configuration according to the Metropolis criterion with probability

$$p = \min\left(1, \exp\left(-2\Delta E_i / k_B T\right)\right), \text{ with } \Delta E_i = s_i \sum_j s_j J_{ij}.$$

Using this stochastic approach to generate states results in a chain of (correlated) configurations that, after an equilibration period, become samples from the Boltzmann distribution defined in Eq. 2. Although there exist cluster algorithms that decorrelate quickly at criticality in equilibrium (Luijten & Blöte, 1995; Fukui & Todo, 2009; Flores-Sola et al., 2017), and recent advances for single spin flip simulations (Müller et al., 2023) that avoid exact calculations of $\Delta E_i$ by exploiting the way Monte Carlo simulations are constructed, the calculation of $\Delta E_i$ is central for simulations of many models. **We therefore use the prediction of $\Delta E_i$ as our long-range task**.

To obtain samples from the target distribution for the dataset, we implement the single cluster variant for the LRIM as presented by Flores-Sola et al. (2017). We make sure to equilibrate the simulation before measuring and to decorrelate subsequent samples by a sufficient number of cluster updates. For more details, see Appendix B.

## 4    LRIM Graph Benchmark

**The long-range graph learning task**   Our goal is to directly translate the Ising model into an appropriate graph-based task formulation with controllable long-range interactions while preserving its simplicity. We focus on the $d = 2$ LRIM on a grid lattice and want to predict energy changes throughout the system. Each LRIM instance corresponds to a graph $G$ with $L \times L$ nodes that are arranged in a 2D periodic grid. Note that the topology is shared among all instances and that each node is connected to its 4 nearest neighbors. Moreover, each node has a single feature, representing its physical spin $\{-1, +1\}$. We formulate the energy prediction as a node regression task, where each node $v_i$ has to predict its energy change $\Delta E_i \in \mathbb{R}$. A visual illustration of how the graph task is constructed is shown in Figure 1. An important remark is that learning on a grid graph is a non-trivial challenge for most message-passing architectures. The reason is that a grid graph exhibits an exponential *computational bottleneck* due to its computation tree, as detailed in Arnaiz-Rodriguez & Errica (2025), which can deteriorate performances as shown in previous work (Alon & Yahav, 2021);

Table 1: Overview of all 10 datasets in the LRIM Graph Benchmark. Our benchmark can systematically vary the complexity across 5 graph sizes (256 to 65,536 nodes) and 2 difficulty levels per size controlled by the interaction parameter $\sigma$. Each dataset contains 10,000 spin configurations represented as 2D periodic grid graphs with 4-regular connectivity. The proposed task considers node-level regression to predict energy changes $\Delta E_i$, with performance measured using $\log_{10}$ MSE.

| Dataset | $\sigma$ easy | $\sigma$ hard | Graphs | Nodes | Edges | Avg. Eff. Resistance | Avg. Short. Path | Diameter |
|---|---|---|---|---|---|---|---|---|
| LRIM-16 | 1.5 | 0.6 | 10,000 | 256 | 512 | 0.49 | 8.03 | 16 |
| LRIM-32 | 1.5 | 0.6 | 10,000 | 1,024 | 2,048 | 0.60 | 16.02 | 32 |
| LRIM-64 | 1.5 | 0.6 | 10,000 | 4,096 | 8,192 | 0.71 | 32.01 | 64 |
| LRIM-128 | 1.5 | 0.6 | 10,000 | 16,384 | 32,768 | 0.82 | 64.00 | 128 |
| LRIM-256 | 1.5 | 0.6 | 10,000 | 65,536 | 131,072 | 0.93 | 128.00 | 256 |

**Dataset control and diversity** To thoroughly evaluate the long-range capabilities with the LRIM benchmark across scale and difficulty, we provide five datasets of increasing size and two difficulty levels (easy and hard) for each size configuration based on the values of $\sigma$. In particular, the datasets range from LRIM-16 (256 nodes) to LRIM-256 ($65,536$ nodes). Complete graph and dataset statistics for all variants are provided in Table 1, with further details on the calculation in Appendix C. Each dataset variant contains $10,000$ distinct graph instances. For each size, the long-range interaction strength is varied between a "hard" variant ($\sigma = 0.6$) and an "easy" variant ($\sigma = 1.5$) (see Section 3 for the justification of these values). Across all settings, our task formulation has the advantage of yielding continuous feedback as the true energy can be more closely approximated by integrating more long-range information. Simultaneously, the harder tasks require models to aggregate information from more distant nodes, as we validate later.

**Data generation** Data generation follows the outlined Monte Carlo sampling protocol of Section 3. For each system size $L$ and $\sigma$ value, we first determine the appropriate pseudo-critical temperature $T_c(\sigma, L)$ where the system exhibits longest correlation lengths, thereby creating the most interesting interactions. Each configuration is sampled from a simulation which is first equilibrated followed by a decorrelation phase between two subsequent samples to ensure statistical independence between them. Then, we split the dataset according to $80/10/10$ into dedicated splits for training, validation, and testing. For more details we refer to Appendix B.

**Task evaluation metrics** We want to highlight the computational efficiency aspect when evaluating methods for their long-range capabilities, as known improvements often come with increased computational costs. To ensure rigorous and fair comparisons, we **require that methods report their runtime complexity for their computational budget** (e.g. $\mathcal{O}(L \cdot E)$ for standard MPNNs with $L$ layers and $E$ edges) **and any precomputation costs**, including creation of additional structure and feature preprocessing. We put little restriction on what can be used on the benchmark on purpose, for instance extra features, encouraging novel methods. However, **we ask that all modifications and their computational overhead must be transparently documented**. To report prediction performance, we use the base-10 logarithm of the mean square error (LogMSE).

## 5 EVALUATION

### 5.1 LONG-RANGE ANALYSIS

In this section, we empirically and theoretically demonstrate that our proposed LRIM benchmark is suitable for testing long-range interactions. We remind the reader that the dataset consists of synthetic simulation data over which we have full knowledge and control of the underlying generation process. This allows us to observe the (in)ability of current models to capture long-range dependencies and encourage the design and development of new approaches that solve our benchmark *under computational budgets* as mentioned above. In our analysis, we consider three different perspectives: *i)* how the simulation accuracy degrades when an oracle is restricted to local neighborhoods, including worst-case error bounds; *ii)* the trade-off between overfitting and generalization based on

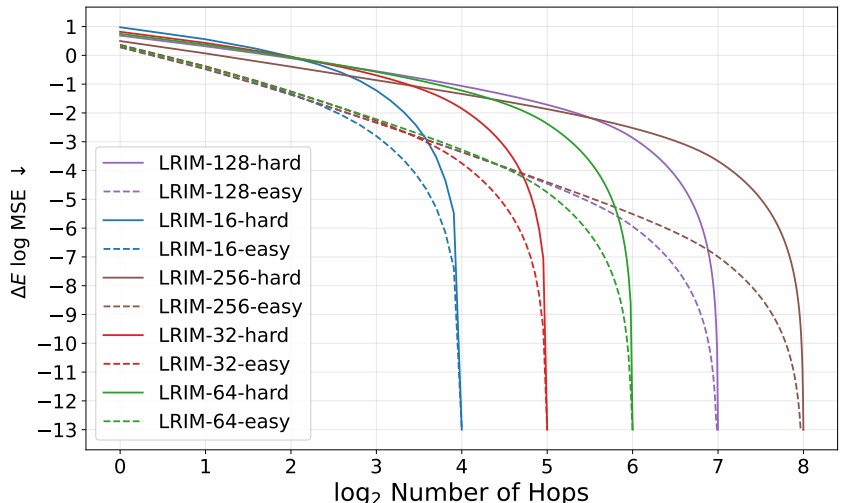

Figure 2: LogMSE ($\downarrow$) performance of the oracle predictor degrades when restricted to consider local r-hop neighborhoods only. The oracle uses the true underlying energy function, but only considers spins within hop-distance r from each target node. Results demonstrate that smaller $\sigma$ values (harder tasks) require larger neighborhoods to achieve the same accuracy, confirming stronger long-range dependencies. Second, larger system sizes increase task difficulty, even within the same $\sigma$. Moreover, the performance decays smoothly, providing a continuous feedback both during evaluation and training. Therefore, achieving low prediction error requires information from neighborhoods spanning significant fractions of the graph, providing model-agnostic evidence that local information is insufficient.

the Weisfeiler-Leman (WL) test of graph isomorphism; *iii)* and theoretical results on recent long-rangedness metrics (Bamberger et al., 2025). Taken together, these analyses provide strong evidence that our benchmark requires long-range reasoning capabilities grounded in well-known physics literature, contrary to previous works.

First, we analyze how prediction accuracy degrades when the information of an oracle is restricted to local r-hop neighborhoods only. The oracle predicts $\Delta E_i$ based on the correct contributions within the r-hop neighborhood; it follows that when the oracle has global access to the graph for each node, its prediction matches the simulation ground truth. Note that this is not a strict lower bound on achievable error for a given r-hop neighborhood.

We vary the parameter $r$ from 1 to the diameter of the graph across our datasets, and plot the oracle's score in Figure 2. These results show that task difficulty can be precisely controlled by both the parameter $\sigma$ and the size of the system $L$. We observe that lower $\sigma$ values consistently require larger neighborhoods to achieve the same prediction accuracy, which is indicative of stronger long-range interactions required to solve the harder task ($\sigma = 0.6$). Furthermore, for the same $\sigma$, a larger system size $L$ also contributes to increase the task's difficulty. The prediction error decays smoothly as the neighborhood size increases from local to global, showing that incorporating information from more distant nodes consistently improves prediction accuracy. This provides both a continuous feedback signal for both model performance evaluation and during training. Crucially, this analysis reveals that achieving low error requires considering interactions across substantial fractions of the entire graph.

Further, we provide theoretical arguments that underscore the fundamental necessity of long-range information for the best performance of the task. We first establish a lower bound on the worst-case error that any method restricted to local neighborhoods must exhibit by not considering the rest of the graph. As part of the system remains unknown, there exists a variety of possible configurations compatible with the observed neighborhood which exhibit a wide range of possible energies. Therefore, no predictor that only considers the local information can have a maximum error significantly below that range. For the detailed derivation we refer to Appendix D.

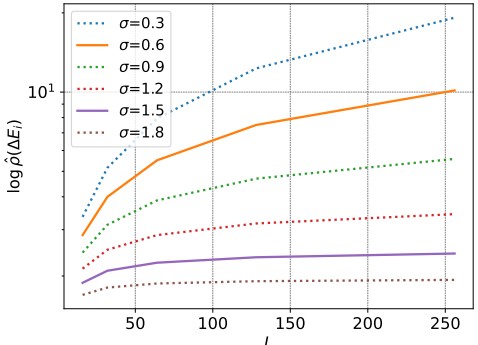

Figure 3: Normalized range measure $\hat{\rho}_i(\Delta E)$ increases with system size $L$ and decreases with $\sigma$. The measure quantifies the relative contribution of distant nodes to the energy prediction task, computed analytically using the oracle's gradient. Smaller $\sigma$ values lead to higher measures across all system sizes, with the growth rate accelerating for $\sigma \leq 1$. This validates both the dependence and controllability of long-range dependencies throughout our proposed benchmark.

**Lemma 5.1.** *Let $f_\theta$ be a model that predicts $\Delta E$ for a given node $v$ on a spin configuration $X$ with a periodic grid graph $G$ of size $N = n \times n$ with diameter $D = n$. If $f_\theta$ only considers the spins of nodes within radius $r \ll D$ of $v$, then there exists a spin configuration $X'$ where $f_\theta(X)_v = f_\theta(X')_v$ but $|Y'_v - f_\theta(X')_v| \geq n^{-\sigma}$.*

Following Leman & Weisfeiler (1968), we analyze limitations of MPNNs and find that WL equivalence classes initially differ in energy targets, requiring larger receptive fields to be distinguished. However, this effect diminishes faster than our oracle analysis due to finite data (detailed analysis in Appendix D). This indicates that expressiveness does not fundamentally restrict fitting the data, however proper generalization likely requires receptive fields closer to the oracle predictor.

**Long-rangedness Measure** We connect our proposed datasets to the long-rangedness metric for graphs defined in Bamberger et al. (2025). The range measure $\rho_u(F)$ of a node-level differentiable function $F$ for a node $u$ and its normalized measure $\hat{\rho}_u(F)$ are defined as

$$\rho_u(F) = \sum_{v \in V} \left| \frac{\partial F(X)_u}{\partial x_v} \right| d_G(u, v), \qquad \hat{\rho}_u(F) = \frac{\rho_u(F)}{\displaystyle\sum_{v \in V} \left| \frac{\partial F(X)_u}{\partial x_v} \right|}.$$

Here, $x_v$ is the node feature of $v$ and $d_G : V \times V \to \mathbb{R}$ is a distance function, on a graph $G = (V, E)$ with nodes $V$ and edges $E$. We adapt the measure to the oracle predictor, which is given by the data generation process, by setting $\Delta E$ as $F$ (assuming spins are embedded in $\mathbb{R}$) and consider $\pm 1$ spins as node features $X$. We first present one of our main results on the expression of the adapted long-range measure in the following proposition. The proof and further derivation of the proposition are provided in Appendix G:

**Proposition 5.2.** *Let $G$ be a periodic long-range Ising model defined over a grid of size $L \times L$, with $F = \Delta E$. Assume that the binary spins $s_i$ are relaxed to continuous spin values $s_i \in \mathbb{R}$. Then, when $d_G : V \times V \to \mathbb{R}$ is the shortest path distance, the range measures $\rho_i(\Delta E)$ and $\hat{\rho}_i(\Delta E)$ are represented as*

$$\rho_i(\Delta E) = \sum_{1 \leq \ell \leq r} \ell \sum_{k \in N_\ell(i)} J_{ik}, \qquad \hat{\rho}_i(\Delta E) = \frac{\rho_i(\Delta E)}{\displaystyle\sum_{1 \leq \ell \leq r} \sum_{k \in N_\ell(i)} J_{ik}},$$

*where $N_\ell(i)$ denotes the set of spins at $\ell$-hop distant from $i$.*

In addition, the following proposition, which states the effect of changing the $\sigma$ parameter on the metric proposed in (Bamberger et al., 2025), supports the controllability of long-rangedness in terms of $\rho_i(\Delta E)$ and $\hat{\rho}_i(\Delta E)$ in LRIM. Its proof is provided in Appendix G for Proposition G.4.

**Proposition 5.3.** *Given the same assumption as Proposition 5.2 with $r = L$, both range measures $\rho_i(\Delta E)$ and $\hat{\rho}_i(\Delta E)$ diverge as $L \to \infty$ when $\sigma \leq 1$, and converge when $\sigma > 1$.*

We plot the normalized range measure $\hat{\rho}_i(\Delta E)$ obtained in Proposition 5.2 across sizes and for various $\sigma$ in order to demonstrate the impact on the long-range measure. The computation of $\hat{\rho}_i(\Delta E)$ relies on an analytical expression of its intermediate expression $\sum_{k \in N_\ell(i)} J_{ik}$, whose formal statement and proof are provided in Theorems G.2 and G.3. We observe in Figure 3 a clear upward trend that smaller $\sigma$ values lead to higher corresponding node value in the metric across different grid sizes.

We also observe that the growth rate of the measure significantly increases for smaller $\sigma$, which suggests that smaller $\sigma$ could increase the likelihood that machine learning models face more difficulty in capturing long-range dependency. We believe the presented result further supplements the legitimacy of using the LRIM as a benchmark for evaluating the performance of models in graph learning focusing on long-range interactions.

Table 2: Baseline performance on LRIM-16 and LRIM-32 datasets. The number of edges E corresponds to 4N in our datasets. We emphasize the importance of reporting computational complexity alongside performance results, as scalability is a crucial aspect in long-range modeling.

| | Preprocessing | Computation | LRIM-16-hard ↓ | LRIM-32-hard ↓ | LRIM-16-easy ↓ | LRIM-32-easy ↓ |
|---|---|---|---|---|---|---|
| GIN | - | $\mathcal{O}(L \cdot E)$ | -2.533 ± 0.313 | -2.249 ± 0.135 | -3.564 ± 0.111 | -3.446 ± 0.307 |
| GatedGCN | - | $\mathcal{O}(L \cdot E)$ | -3.844 ± 0.055 | -4.087 ± 0.238 | -4.817 ± 0.111 | -4.940 ± 0.298 |
| GatedGCN-VN$_G$ | $\mathcal{O}(N)$ | $\mathcal{O}(L \cdot E + L \cdot N)$ | -4.068 ± 0.131 | -3.243 ± 0.177 | -4.612 ± 0.101 | -4.322 ± 0.229 |
| GPS-Base | - | $\mathcal{O}(L \cdot N^2)$ | -4.211 ± 0.155 | -4.044 ± 0.122 | -5.296 ± 0.026 | -5.134 ± 0.054 |
| GPS-RWSE | $\mathcal{O}(k \cdot N^2)$ | $\mathcal{O}(L \cdot N^2)$ | -4.011 ± 0.129 | -4.134 ± 0.075 | -5.133 ± 0.235 | -5.103 ± 0.091 |
| GPS-LapPE | $\mathcal{O}(k^2 \cdot E)$ | $\mathcal{O}(L \cdot N^2)$ | -4.334 ± 0.065 | -4.032 ± 0.092 | -5.154 ± 0.177 | -4.858 ± 0.231 |

## 5.2 EMPIRICAL EVALUATION

The last part of the section presents our quantitative results. We evaluate two of the most common and popular message-passing architectures, namely GIN (Xu et al., 2019) and GatedGCN (Bresson & Laurent, 2018). In addition, we include a MPNN variation using a virtual node (Southern et al., 2025) and three versions of GPS (Rampášek et al., 2022) as representatives of graph transformer models: one that augments node features with random-walk structural encodings (RWSE); one that relies on Laplacian positional encodings (LapPE) instead; and another that has no additional information (Base). To provide some initial results, that practitioners can compare their methods against, we carefully train and evaluate these models on the easy and hard variants of LRIM-16 and LRIM-32. To ensure a rigorous evaluation, we perform model selection on the most important hyperparameters: the model readout, hidden dimension, and number of layers. The number of layers proved to be the most impactful parameter, as intuitively expected, which we chose as a relative percentage of the system size. For more details on the model selection and training of the baselines we refer to Appendix E. We report the mean test LogMSE in Table 2 with standard deviations computed over 3 runs. On LRIM-16, we observe a consistent difference between the easy and hard variants. The message-passing based models, which require substantially less computation than graph transformers, perform worse. This highlights the importance of comparing models with respect to their computational complexity, as this can significantly impact the results and incentivize future work on the trade-offs between computational requirements and performance.

On the larger datasets, we expect the situation to be worse. In this setup, due to computational constraints, we do not directly train models but rather evaluate the transfer ability of LRIM-16-hard baselines on the larger hard versions of the benchmark: LRIM-32, LRIM-64, LRIM-128 and LRIM-256. These transferability results are reported in Table 3. We note that there is a smooth increase in the test error as we evaluate all models on systems of larger sizes, as expected. The LogMSE of the message-passing architectures tends to saturate, possibly because the relative error is quite high. Using a naive implementation of the attention, which requires quadratic memory, one cannot perform inference on the largest instances even with a batch size 1 on an A100 80GB GPU, as it would require around 160GB of VRAM. If we compare these results with the oracle predictions of Figure 2, it is clear that the graph community still has a long path ahead in designing novel models

Table 3: We evaluate how well models trained on LRIM-16-hard transfer to larger systems without additional training. The results show that performance generally degrades as system size increases. GPS variants encounter out-of-memory (OOM) errors on LRIM-256 even for inference-only, demonstrating the importance of considering the scalability of long-range methods.

| | Preprocessing | Computation | LRIM-16-hard ↓ | LRIM-32-hard ↓ | LRIM-64-hard ↓ | LRIM-128-hard ↓ | LRIM-256-hard ↓ |
|---|---|---|---|---|---|---|---|
| GIN | - | $\mathcal{O}(L \cdot E)$ | $-2.406 \pm_{0.148}$ | $-1.043 \pm_{0.051}$ | $-0.774 \pm_{0.042}$ | $-0.703 \pm_{0.041}$ | $-0.903 \pm_{0.047}$ |
| GatedGCN | - | $\mathcal{O}(L \cdot E)$ | $-3.919 \pm_{0.223}$ | $-1.050 \pm_{0.004}$ | $-0.781 \pm_{0.002}$ | $-0.708 \pm_{0.003}$ | $-0.952 \pm_{0.005}$ |
| GatedGCN-VN$_G$ | $\mathcal{O}(N)$ | $\mathcal{O}(L \cdot E + L \cdot N)$ | $-3.756 \pm_{0.063}$ | $-1.054 \pm_{0.006}$ | $-0.788 \pm_{0.004}$ | $-0.716 \pm_{0.005}$ | $-0.968 \pm_{0.008}$ |
| GPS-Base | - | $\mathcal{O}(L \cdot N^2)$ | $-4.340 \pm_{0.101}$ | $-1.057 \pm_{0.000}$ | $-0.790 \pm_{0.001}$ | $-0.719 \pm_{0.001}$ | OOM |
| GPS-RWSE | $\mathcal{O}(k \cdot N^2)$ | $\mathcal{O}(L \cdot N^2)$ | $-4.345 \pm_{0.065}$ | $-1.057 \pm_{0.005}$ | $-0.790 \pm_{0.004}$ | $-0.716 \pm_{0.002}$ | OOM |
| GPS-LapPE | $\mathcal{O}(k^2 \cdot E)$ | $\mathcal{O}(L \cdot N^2)$ | $-4.248 \pm_{0.110}$ | $-1.053 \pm_{0.006}$ | $-0.785 \pm_{0.005}$ | $-0.716 \pm_{0.005}$ | OOM |

that efficiently and effectively capture long-range interactions, even on relatively regular graphs such as the grid.

To investigate the impact of the receptive field, we perform an ablation study on the number of layers across all architectures. In Figure 4 we reuse the best configurations from Table 2 and retrain the models by using between 1 to 32 layers. Note that the datasets have a diameter of 16 and 32 respectively, therefore, we would expect improvement up to that point. We observe that performance initially consistently improves with increased depth. However, the performance flattens out after roughly 10 and 16 layers, respectively, on the different system sizes. While GT baselines seem to exhibit a slight edge compared to the MPNN baselines. They see the whole graph immediately due to positional encodings and attention mechanisms. For the message-passing architectures, this may be partly due to well-known problems such as computational bottlenecks and vanishing gradients, which hamper the ability of these architectures to properly learn a good representation. For the graph transformers, it may be tied to a too unrestricted attention mechanism. We provide an additional ablation investigating scaling the number of parameters used for the GatedGCN baseline in Appendix E.4.

Notwithstanding these considerations, what we want to stress here is how our LRIM benchmark has uncovered, in a rather explicit way, that both message-passing and graph transformer architectures suffer from limitations when modeling this kind of long-range problems, and that our contribution stands as a novel, theoretically, and empirically grounded tool towards developing and evaluating more capable long-range techniques for graph learning.

## 6 LIMITATIONS

Our proposed LRIM benchmark is a synthetic dataset with a well-understood and well-defined task, and as such it cannot be directly tied to a real-world problem. Its primary purpose and main advantage, as argued throughout the paper, is to provide an understandable and controllable framework to assess long-range capabilities rather than to solve an open problem. As such, we do not intend LRIM as a replacement for real-world benchmarks, but rather as a complementary tool for advancing the study of long-range interactions for graph learning. Furthermore, our current benchmark is limited to regular lattice structures, which may not capture the diverse topological patterns encountered in general graphs. Future work may incorporate other structured graph types, but this would require careful consideration of how to properly obtain appropriate simulated data (particularly, finding the critical temperature for such systems) and the accompanying long-range analysis.

## 7 CONCLUSION

With current graph learning benchmarks, it is difficult to properly isolate and assess the ability of models to capture long-range dependencies. To address these ambiguities, we introduced the Long-Range Ising Model (LRIM) Graph Benchmark, a physics-based benchmark based on the power-law interacting Ising model that relies on controllable and provable long-range dependencies. We can precisely vary the hardness of the task across datasets, which scale from 256 to 65k nodes. In addition, we provide model-agnostic evidence that LRIM tasks genuinely require long-range reasoning, with oracle prediction degrading when information is restricted to local neighborhoods. This is further supported through theoretical insights using recent long-rangedness measures. Our empir-

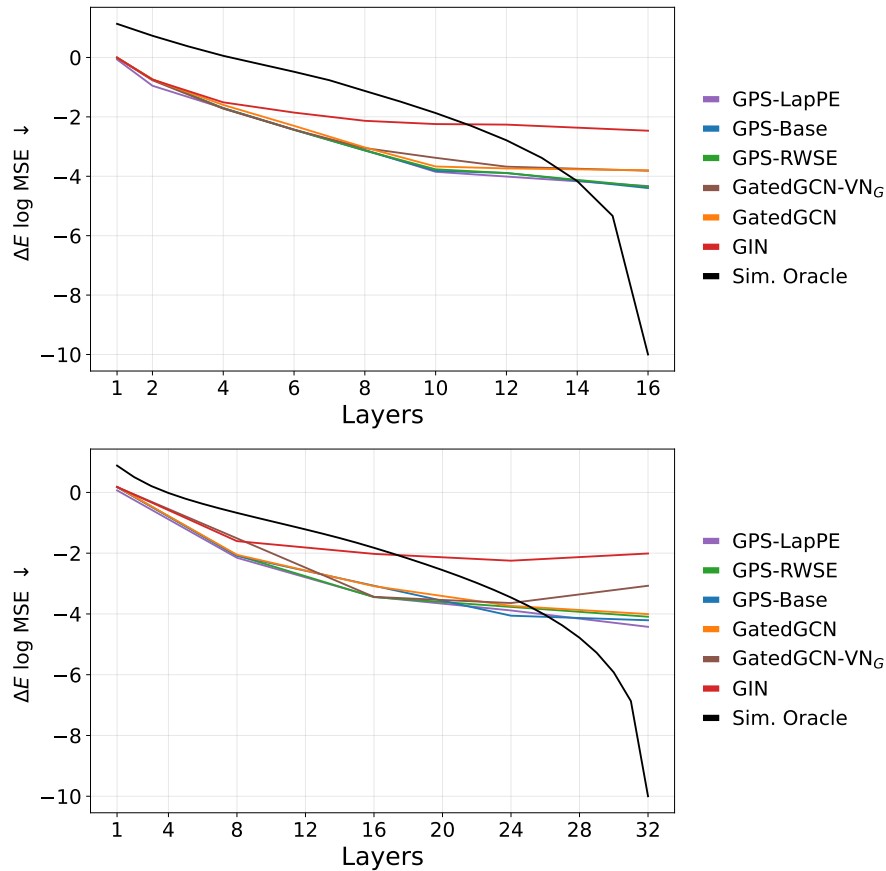

Figure 4: LogMSE (↓) of the layer ablation study plotting performance as a function of number of layers. Oracle performance is clamped at -10 for visualization purposes. Note, each combination of model and layer is a separate trained model instance. On the top, models are trained and evaluated on LRIM-16-hard, while on the bottom they are trained and evaluated on LRIM-32-hard. All models consistently improve with increased depth but plateau with a significant gap remaining to the oracle predictor.

ical evaluation reveals large gaps between current methods and oracle performance, highlighting limitations in existing graph learning approaches when confronted with provably long-range tasks. Our aim is that the benchmark will contribute to establishing a foundation for developing, properly evaluating, and advancing our understanding of what architectural innovations are needed to tackle long-range dependency modeling in graph-structured data.

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

# A  ORACLE PREDICTOR

## A.1  ALL

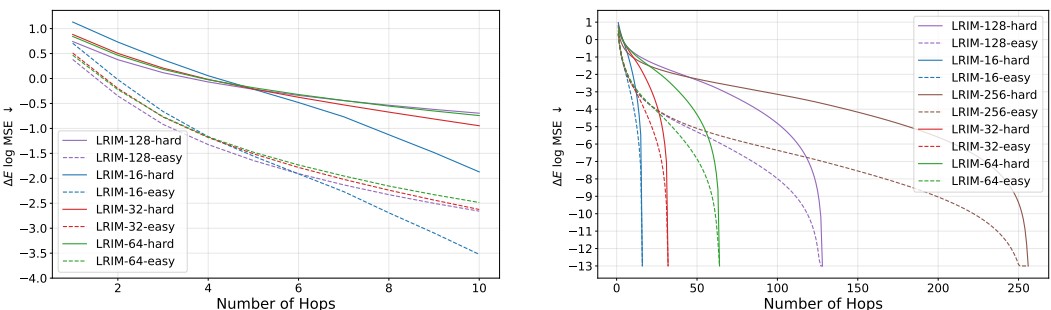

Figure 5: LogMSE (↓) performance of the oracle predictor across **all** LRIM datasets when restricted to r-hop neighborhoods. (Left) indicates the first 10 hops, while (right) shows performance across the graph diameter. Performance degrades smoothly as neighborhood size decreases. Further, harder variants consistently require larger neighborhoods than easier variants for the same accuracy. Finally, larger sizes also increase task difficulty even for fixed $\sigma$. The oracle uses the true underlying Ising energy function but only considers spins within hop-distance $r$, providing model-agnostic evidence that our benchmark tasks genuinely require long-range information across substantial parts of the graph.

## A.2 DOUBLE

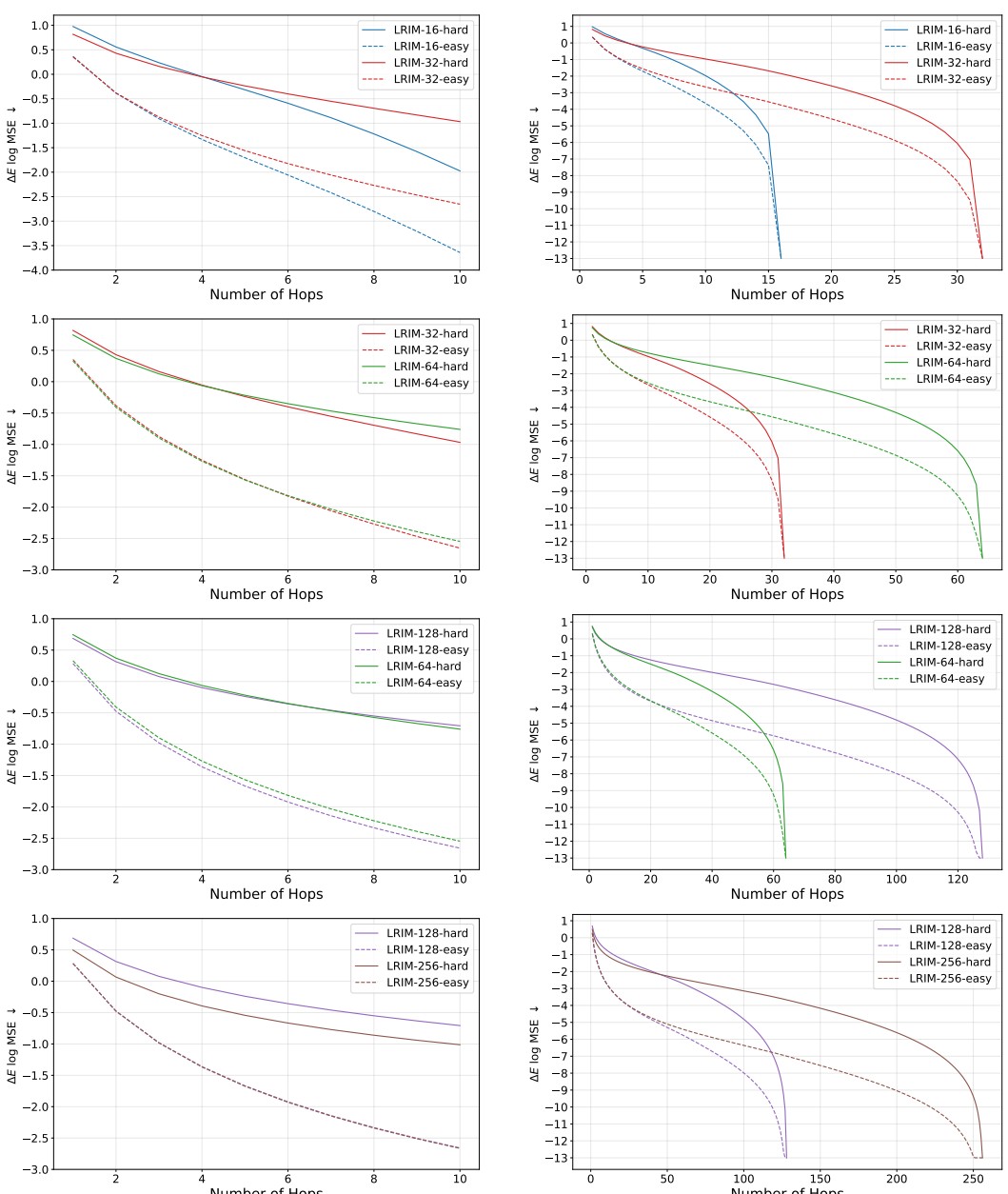

Figure 6: LogMSE (↓) performance of the oracle predictor across size **adjacent** LRIM datasets when restricted to r-hop neighborhoods. (Left) indicates the first 10 hops, while (right) shows performance across the graph diameter. Performance degrades smoothly as neighborhood size decreases. Further, harder variants consistently require larger neighborhoods than easier variants for the same accuracy. Finally, larger sizes also increase task difficulty even for fixed $\sigma$. The oracle uses the true underlying Ising energy function but only considers spins within hop-distance $r$, providing model-agnostic evidence that our benchmark tasks genuinely require long-range information across substantial parts of the graph.

## A.3 SINGLE

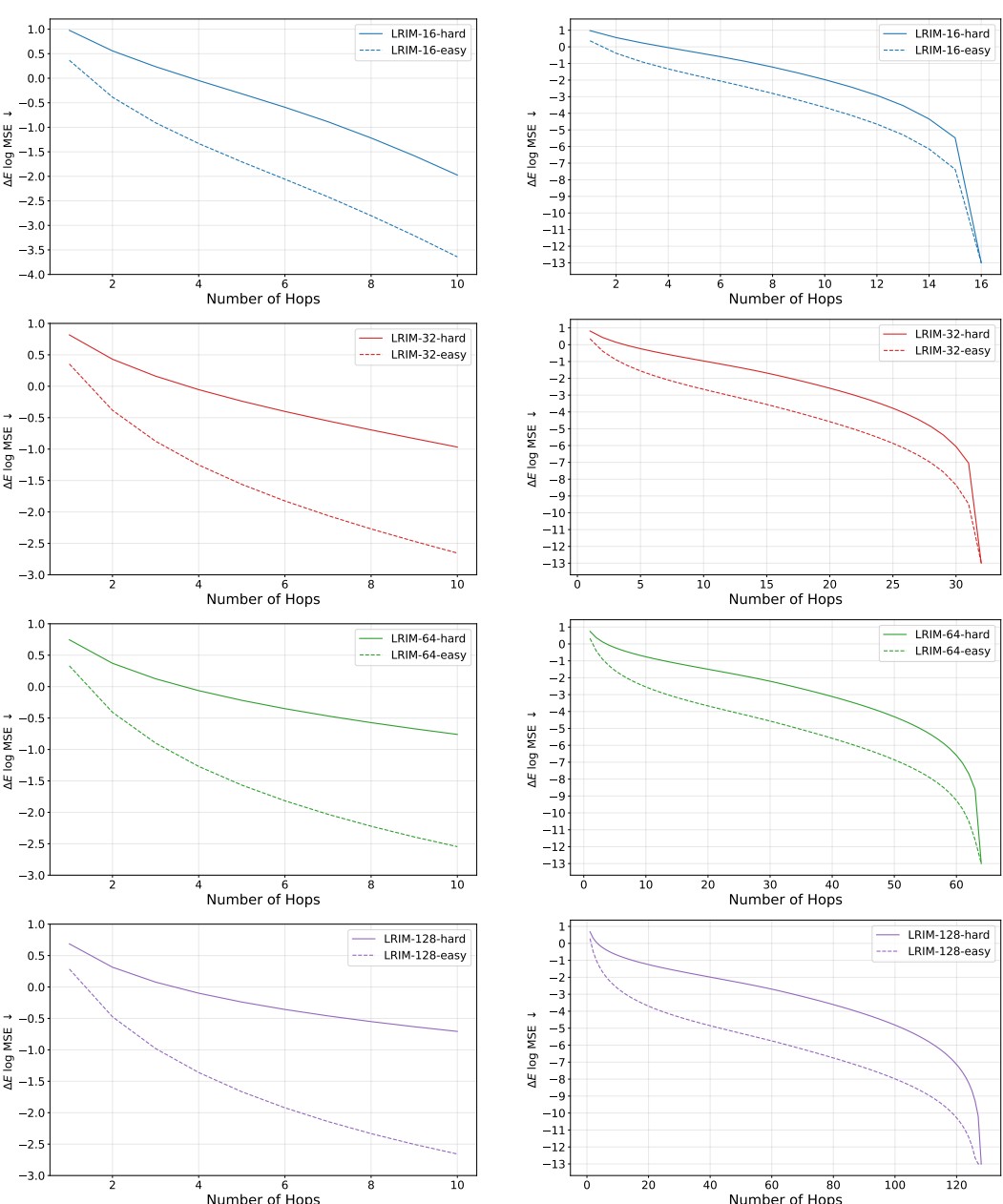

Figure 7: LogMSE (↓) performance of the oracle predictor across a **single** LRIM dataset when restricted to r-hop neighborhoods. (Left) indicates the first 10 hops, while (right) shows performance across the graph diameter. Performance degrades smoothly as neighborhood size decreases. Further, harder variants consistently require larger neighborhoods than easier variants for the same accuracy. Finally, larger sizes also increase task difficulty even for fixed $\sigma$. The oracle uses the true underlying Ising energy function but only considers spins within hop-distance $r$, providing model-agnostic evidence that our benchmark tasks genuinely require long-range information across substantial parts of the graph.

## B  ISING MODEL: BACKGROUND AND SIMULATION DETAILS

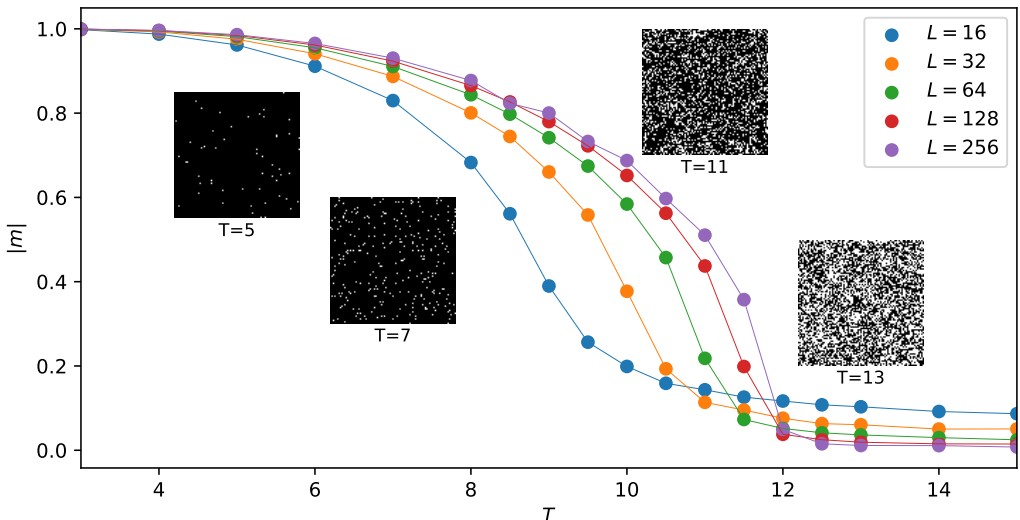

Figure 8: Absolute of the magnetic order parameter per spin $|m| = 1/L^2 \sum_i \mathbf{s}_i$ versus temperature $T$ for the LRIM with $\sigma = 0.6$ for different system sizes $L$ as noted in the figure. The snapshots show respective configurations from simulations for $L = 256$ at the mentioned temperatures. The plot shows the transition from ordered configurations at small $T$ to disordered configurations at large $T$ and the associated shift in the transition temperature for different system sizes. The pseudo-critical temperature $T_c$ marks the temperature at which the spins are highly correlated.

The (long-range) Ising model has three distinct phases depending on the temperature $T$; it is disordered above $T_c$, critical at $T_c$, and ordered below $T_c$. The transition from disordered to ordered phase is evident from the behavior of the magnetic order parameter $m = \sum_i s_i/N$, defined as average spin orientation of the system. In Figure 8 we plot the absolute magnetization $|m|$ as function of temperature $T$ for the LRIM with $\sigma = 0.6$ for various system sizes $L$. At low temperatures, the system has close to unit magnetization and is ordered apart from very few thermal excitations. This is reconfirmed by the snapshots of the system configuration for $T = 5$ and $T = 7$ in the same figure. For large temperatures, $|m|$ is approaching zero, corresponding to a random or *disordered* configuration.

Between these two phases there is a transition, characterized by so-called *critical* behavior with diverging correlations in the system. For our graph benchmark, we determine this point of transition by determining the pseudo-critical temperature for each system size. As discussed in the main text, this temperature is used for the benchmark, as it guarantees correlation between spins, as discussed in the main text.

### B.1  DETERMINATION OF THE PSEUDO CRITICAL TEMPERATURE

We determine the pseudo-critical temperature from the location of maxima of the magnetic susceptibility, defined as

$$\chi_m = \frac{\partial m}{\partial \beta} = k_b T N \left( \langle m^2 \rangle_T - \langle m \rangle_T^2 \right), \tag{3}$$

where $N = L \times L$ is the number of spins, $m = \sum_i s_i/N$ is the magnetization per spin, and $\langle \ldots \rangle$ symbolizes the expectation under the Boltzmann distribution at a given temperature $T$. The magnetic susceptibility is related to the correlation function as

$$\chi_m = k_b T \sum_{\mathbf{r}} C(\mathbf{r}), \tag{4}$$

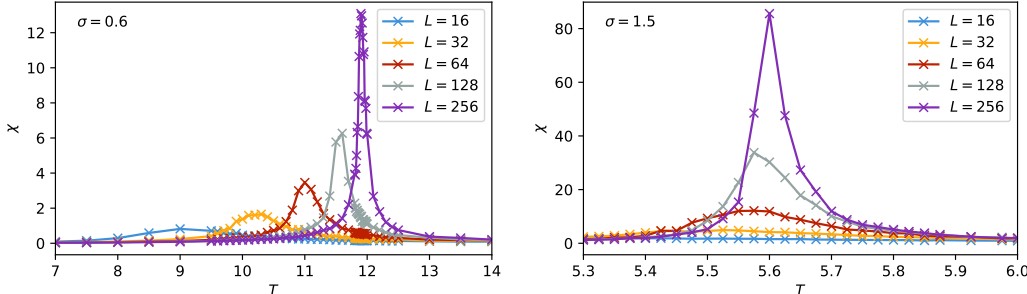

Figure 9: Magnetic susceptibility $\chi = k_b T N \left( \langle m^2 \rangle - \langle |m| \rangle^2 \right)$ as function of temperature $T$ for the long-range Ising model for $\sigma = 0.6$ (left) and $\sigma = 1.5$ (right). The positions of the maxima of $\chi$ mark the position of pseudo $T_c$, and are used to extract our simulation temperature as discussed in Appendix B.1. For $\sigma = 0.6$, the slower decaying system, the system size $L$ influences the pseudo critical temperature much more strongly than for the more short-range model with $\sigma = 1.5$.

which means that the maximum of $\chi$ also marks the maximum of the system-wide (summed) connected correlation function over all distance vectors $\mathbf{r}$. In practice, as usual for finite systems, we consider subtracting the expectation of the absolute magnetization $|m|$, resulting in

$$\chi = k_b T N \left( \langle m^2 \rangle_T - \langle |m| \rangle_T^2 \right). \tag{5}$$

This avoids symmetry cancellations due to the simulation of finite systems with a cluster algorithm. The peaks of $\chi_m$ and $\chi$ coincide in the absence of external magnetic field, thus the maximum of $\chi$ corresponds to maximizing the correlations in the system.

To locate the maximum of $\chi$, we perform a fit of a quadratic function to the maximum and surrounding data points. Standard errors of pseudo $T_c$ are estimated by leave one out jackknife resampling (Efron, 1992).

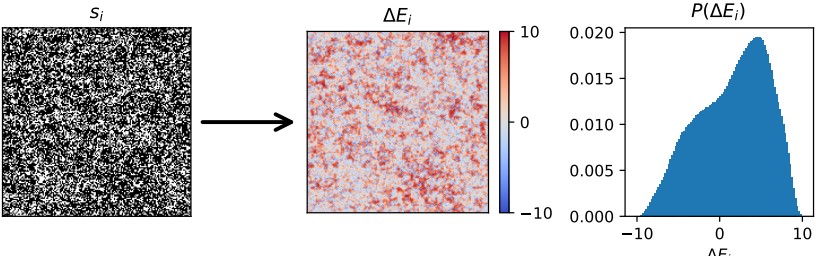

Figure 10: Visualization of the mapping from spin configuration $\mathbf{s}_i$ (left) to the corresponding $\Delta E_i$ (middle); (right) the resulting histogram $P(\Delta E_i)$ of $\Delta E_i$. The presented representative snapshot was generated for $\sigma = 0.6$ and $L = 256$ at the pseudo critical point.

## B.2 MARKOV CHAIN MONTE CARLO SIMULATIONS

A default approach to generate samples from the target distribution for spin systems in statistical physics is the use of Markov chain Monte Carlo simulations. For this, one (randomly) proposes a change to the system and accepts the proposal in such a way that the stationary distribution of the samples follows the target distribution. Specifically, for the LRIM, one proposes a spin flip at a random location $i$, and accepts the proposal with Metropolis probability given by

$$p = \min \left( 1, \exp \left( -2\Delta E_i / k_B T \right) \right), \text{ with } \Delta E_i = s_i \sum_j s_j J_{ij}.$$

Calculating $\Delta E_i$ involves $N$ terms, where $N$ is the number of spins. Thus, a sweep consisting of $N$ updates, has complexity $O(N^2)$. This is in contrast to short-range interacting spin models,

for which a sweep has complexity $O(N)$. Pseudo code implementation of a standard Metropolis MCMC simulation for the LRIM is shown in Algorithms 1 and 2. Figure 10 demonstrates the mapping from spin configuration $s_i$ to $\Delta E_i$ for a single snapshot of the system. The last plot shows the normalized histogram of $P(\Delta E_i)$ of energy difference values at the critical point, which has a clearly non-trivial non-Gaussian shape.

---

**Algorithm 1:** DeltaEnergyAtSite(J, s, x, y)

---

**Input:** Interaction strength $J \in \mathbb{R}^{L \times L}$, spins $s \in \{-1, +1\}^{L \times L}$, site $(x, y)$
$h \leftarrow 0$;
**for** $u = 0$ **to** $L - 1$ **do**
    **for** $v = 0$ **to** $L - 1$ **do**
        $\Delta x \leftarrow \min(|x - u|, L - |x - u|)$, $\Delta y \leftarrow \min(|y - v|, L - |y - v|)$;
        $h \mathrel{+}= J[\Delta x, \Delta y] \cdot s[u, v]$;
$\Delta E \leftarrow 2 \cdot s[x, y] \cdot h$;
**return** $\Delta E$;

---

**Algorithm 2:** MetropolisMonteCarlo

---

**Input:** $J, \beta$, number of samples $N$, RNG seed $R$
Initialize RNG with $R$; initialize $s[i, j] = \pm 1$ randomly;
**for** $k = 1$ **to** $N$ **do**
    **for** $t = 1$ **to** $L^2$ **do**
        Draw $(x, y)$ uniformly from $\{0, \ldots, L - 1\}^2$;
        $\Delta E \leftarrow$ DeltaEnergyAtSite$(J, s, x, y)$;
        $\alpha \leftarrow \min(1, \exp(-\beta \, \Delta E))$;
        Draw $r \sim \text{Unif}[0, 1]$;
        **if** $r < \alpha$ **then** set $s[x, y] \leftarrow -s[x, y]$;

---

Such local updates for the LRIM are thus very costly to perform since all spins interact with each other. In addition, at the critical point, one has critical slowing down, i.e., the autocorrelation times grow strongly, often exponentially. This is due to the divergence of the physical correlation length that slow modes of the Markov chain must decorrelate across. Non-local cluster updates that generate configurations that decorrelate quickly at the critical point are a hallmark of statistical physics for simulating short-range (Swendsen & Wang, 1987) and long-range (ferromagnetic) spin models (Luijten & Blöte, 1995). We here utilize the formulation of Flores-Sola et al. (2017) to simulate the LRIM. A single cluster update then consists of the following steps:

1. A spin $s_i$ is chosen at random as the initial seed and put on a stack. Flip the spin orientation $s_i \rightarrow -s_i$.

2. If the stack is not empty, pop the top element; otherwise, go to step 1) to construct the next cluster.

3. Generate a number of bonds $n_i$ to be considered from a Poisson distribution $\text{Pois}(2\lambda_i)$ with $\lambda_i = \sum_{j \neq i} J_{ij}/k_b T$.

4. Pick the $n_i$ bonds categorically according to their probability $p_j = \lambda_{ij}/\lambda_i$, where $\lambda_{ij} = 2J_{ij}/k_b T$. This can be done in $O(1)$ using the Alias-Walker method (Walker, 1974).

5. If spin $s_j \neq s_i$, put $s_j$ on the stack and flip $s_j$.

6. Go to step 2).

We also provide pseudo code implementations of this in Algorithms 3, 4, and 5. Algorithm 3 the pseudo code to construct the Alias-Walker method is shown, and Algorithm 4 demonstrates how to sample an event according to its probability from it. Algorithm 5 demonstrates the implementation of the full cluster based simulation.

---

**Algorithm 3:** BuildAlias(J)

---

**Input:** Interaction strength $J \in \mathbb{R}^{L \times L}$
**Output:** Alias–Walker tables (alias, prob) over $L^2$ offsets; $\lambda = \sum_{i,j} J[i,j]$

$\lambda \leftarrow \sum_{i=0}^{L-1} \sum_{j=0}^{L-1} J[i,j]$;

Flatten offsets $(i,j)$ to indices $u \in \{0, \ldots, L^2 - 1\}$ and set $p_u \leftarrow \frac{J[i,j]}{\lambda} \cdot L^2$;

Initialize two worklists: $S \leftarrow \{u : p_u \leq 1\}$, $L \leftarrow \{u : p_u > 1\}$;

**while** *S and L not empty* **do**

    Pop $u \in S$ and $v \in L$;

    prob[u] $\leftarrow p_u$; alias[u] $\leftarrow v$;

    $p_v \leftarrow p_v - (1 - p_u)$;

    **if** $p_v \leq 1$ **then** move $v$ to $S$;

    **else** keep $v$ in $L$;

**return** (alias, prob, $\lambda$);

---

**Algorithm 4:** AliasSample(alias, prob)

---

Draw $u \sim \text{Unif}\{0, \ldots, L^2 - 1\}$, $r \sim \text{Unif}[0, 1]$;

**if** $r \leq \text{prob}[u]$ **then** w $\leftarrow u$;

**else** w $\leftarrow \text{alias}[u]$;

Map $w$ back to offset $(\Delta x, \Delta y)$ on $\{0, \ldots, L-1\}^2$;

**return** $(\Delta x, \Delta y)$;

---

**Algorithm 5:** ClusterMonteCarlo

---

**Input:** Interaction strength $J$, inverse temperature $\beta$, number of desired samples $N$, random seed $R$

Initialize RNG with $R$; initialize spins $s[i,j]$ randomly to $\pm 1$;

(alias, prob, $\lambda$) $\leftarrow$ BuildAlias(J);

**for** $t = 1$ **to** samples **do**

    Draw seed location $(x_0, y_0)$ uniformly from $\{0, \ldots, L-1\}^2$;

    Flip spin: $s[x_0, y_0] \leftarrow -s[x_0, y_0]$;

    Initialize stack $S \leftarrow [(x_0, y_0)]$;

    **while** $S \neq \emptyset$ **do**

        Pop $r = (x, y)$ from $S$;

        Draw $n \sim \text{Poisson}(\lambda_\beta)$;

        **for** $e = 1$ **to** $n$ **do**

            $(\Delta x, \Delta y) \leftarrow \text{AliasSample}(\text{alias}, \text{prob})$;

            $(u, v) \leftarrow ((x + \Delta x) \bmod L, (y + \Delta y) \bmod L)$;

            **if** $s[x_0, y_0] \neq s[u, v]$ **then**

                Push $(u, v)$ to $S$;

                $s[u, v] \leftarrow -s[u, v]$;

## C    DATASET STATISTICS

The diameter

$$D = \max_{u,v \in V(G)} d(u,v)$$

of a graph $G$ is defined as the maximum shortest hop distance between any two nodes in the graph. For a periodic grid graph of size $N$, the distances can be calculated using the manhattan distance and the diameter is $\sqrt{N}$.

The average shortest path of a graph is the average shortest hop distance between two nodes in the graph. For a periodic grid graph of size $N = n \cdot n$ it is given as $\frac{n^3}{2(n^2-1)}$.

$$
\begin{aligned}
SP &= \frac{1}{N(N-1)} \sum_{u \in V(G)} \sum_{v \in V(G), v \neq u} d(u,v) \\
&= \frac{1}{N(N-1)} N \sum_{v \in V(G)} d(u, v_0) && \text{topology of all nodes is the same} \\
&= \frac{1}{(N-1)} \sum_{v \in V(G)} \min\{u_x, n - u_x\} + \min\{u_y, n - u_y\} && \text{manhatten distance} \\
&= \frac{1}{(N-1)} 2 \sum_{v \in V(G)} \min\{u_x, n - u_x\} && \text{symmetry and linearity of coordinates} \\
&= \frac{1}{(N-1)} 2n \sum_{i=1}^{n} \min\{i, n - i\} && \text{repeated summation of each row} \\
&= \frac{1}{(N-1)} 2n \left( \left( 2 \sum_{i=1}^{\frac{n}{2}} i \right) - \frac{n}{2} \right) \\
&= \frac{1}{(N-1)} 2n \left( \frac{n}{2}(\frac{n}{2} + 1) - \frac{n}{2} \right) \\
&= \frac{n^3}{2(n^2 - 1)}
\end{aligned}
$$

The effective resistance was calculated using the networkx implementation following the Kirchhoff index(Ellens et al., 2011) and normalized by the number of edges.

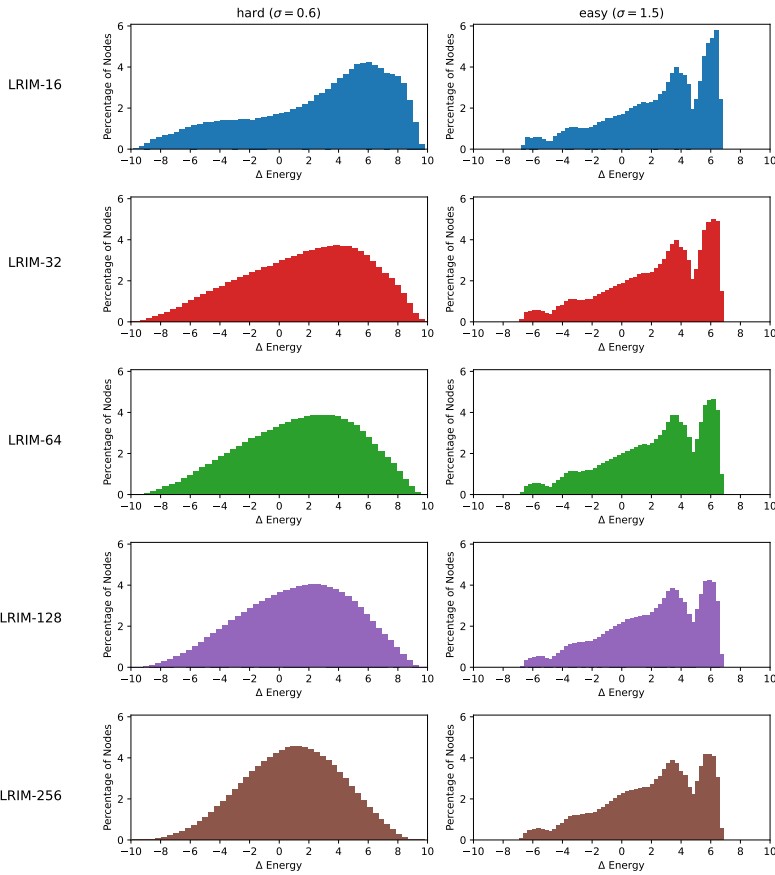

Figure 11: Histogram of the ground-truth $\Delta$ Energies across the various LRIM datasets (first 1000 samples), binned into 50 bins. From top to bottom are the different system sizes 16,32,64,128 and 256.

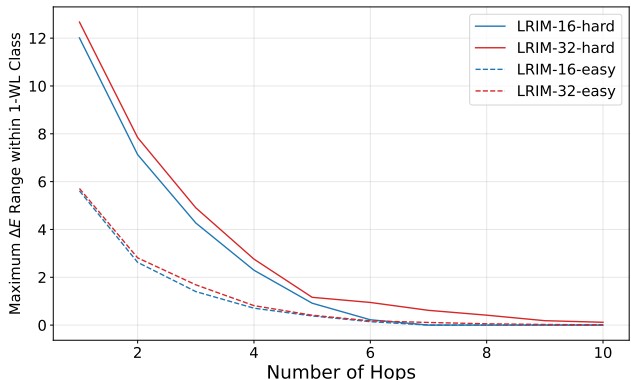

Figure 12: Maximum range of $\Delta E$ values among nodes sharing the same 1-WL equivalence class when restricted to r-hop neighborhoods. Nodes that are indistinguishable to message-passing neural networks cam initially exhibit vastly different energy targets, requiring expanded receptive fields to resolve these ambiguities. However, they decay much faster than the oracle predictor (Figure 2) indicates potential overfitting with finite data, though generalization likely requires distances closer to the oracle. However, this also means that inherently, 1-WL in distinguishability should not pose a major obstacle for MPNNs on our Benchmark.

## D    EVALUATION

In this section, we discuss how easy it would be for MPNNs to fit the training data. The Weisfeiler-Leman (WL) (Leman & Weisfeiler, 1968) test of graph isomorphism provides a theoretical framework to understand the expressive power of MPNNs. Since classical MPNNs are known to be equivalent to the 1-WL test, we use the test as a proxy for MPNNs' fitting abilities. By this test, nodes within the same WL equivalence class must be associated with the same outputs, hence they cannot be discerned by one another. We compute 1-WL labels for all nodes in our datasets up to iteration $k$, creating equivalence classes of nodes that are indistinguishable to $k$-layer MPNNs. For each equivalence class, we measure the range of $\Delta E$ values between the nodes, which indicates if nodes with very different energies are clustered together. Figure 12 shows the maximum range among all equivalent classes depending on the size of the considered neighborhood. As we can see, the maximum range is not negligible initially, but decreases fast as the neighborhood increases. This results highlights two important insights: First, there are nodes with similar neighborhoods that have very different prediction targets in our datasets. This is desirable as it requires information beyond the immediate neighbors to distinguish these cases. As a consequence, there is an inherent drive towards an increased receptive field, which is also necessary to capture long-range dependencies, in order to uniquely shatter the equivalence classes. Second, the curve drops off faster than the analysis of the oracle predictor. That is, because of the finiteness of datasets, there exists a potential shortcut to approximate the true training prediction with fewer than the minimum required number of layers. However, it is crucial to note that this analysis does not provide conclusions about how well models can generalize beyond the training data. In fact, we have shown that the number of layers depends on $\sigma$ and loosely follows the oracle predictor. We want to make practitioners well-aware of this discrepancy between the number of layers required for fitting the training data and those required to approximate the true target function.

**Lemma D.1.** *Let $f_\theta$ be a model that predicts $\Delta E$ for a given node $v$ on an instance $X$ with a periodic grid graph $G$ of size $N = n \times n$ with diameter $D = n$. If $f_\theta$ only considers the spins of nodes within radius $r \ll D$, then there exists a configuration $X'$ where $f_\theta(X)_v = f_\theta(X')_v$ but $|Y'_v - f_\theta(X')_v| \geq n^{-\sigma}$.*

*Proof.* We construct two candidate instances $X'_1, X'_2$, which have the exact same spins as $X$ within radius $r$ and are all $-1$, respectively $+1$ outside of that. The error of any prediction will then be at

least $\frac{1}{2}|Y'_{1,v} - Y'_{2,v}|$.

$$
\begin{aligned}
Y'_{1,v} - Y'_{2,v} &= \sum_{u \in G} x'_{1,u} x'_{1,v} d(u,v)^{-(2+\sigma)} \\
&\quad - \sum_{u \in G} x'_{2,u} x'_{2,v} d(u,v)^{-(2+\sigma)} \qquad \text{def. of } \Delta E \\
&= \sum_{u \in G, d(u,v) > r} x'_{1,u} x'_{1,v} d(u,v)^{-(2+\sigma)} \\
&\quad - \sum_{u \in G, d(u,v) > r} x'_{2,u} x'_{2,v} d(u,v)^{-(2+\sigma)} \\
&= \sum_{u \in G, d(u,v) > r} (x'_{1,u} - x'_{2,u}) x'_{1,v} d(u,v)^{-(2+\sigma)} \\
&= 2 x'_{1,v} \sum_{u \in G, d(u,v) > r} d(u,v)^{-(2+\sigma)}
\end{aligned}
$$

$$
\begin{aligned}
\frac{1}{2}|Y'_{1,v} - Y'_{2,v}| &= \sum_{u \in G, d(u,v) > r} d(u,v)^{-(2+\sigma)} \\
&\geq (N - r^2) \frac{1}{n}^{(2+\sigma)} \\
&\geq n^{-\sigma} - \frac{r^2}{n^{2+\sigma}} \\
&\geq n^{-\sigma} \qquad\qquad r \ll n
\end{aligned}
$$

$\square$

# E  EMPIRICAL EVALUATION

## E.1  CODE AND TRAINING

All our experiments are implemented using pytorch and the pytorch geometric library (Fey et al., 2025). Our code builds on top of a commmon codebase originated on top of graphgym and further used by GraphGPS(Rampášek et al., 2022), GRIT(Ma et al., 2023) and Benchmarking Positional Encodings (Grötschla et al., 2024). The experiments were conducted on a variety of GPUs including: RTX 3090 (24GB VRAM), RTX 2080 (12 GB) and Titan RTX (24GB). Furthermore, we used an A100 GPU with 80GB of VRAM for the inference transferability ablation. Unless stated otherwise, the reported model performances are taken over three different model seeds and model selection is done for each combination of model architecture and dataset.

All models are trained using the MSE loss for at most 2000 epochs using the AdamW (Loshchilov & Hutter, 2017) optimizer with an early stopping of 200 epochs. Further, we limit the compute budget of each model to be at most 48 hours. We use a weight decay of $10^{-5}$ and a cosine scheduler with 5 warmup epochs and gradient clipping (l2 norm of 1).

## E.2  MODEL SELECTION

We follow the systematic hyperparameter optimization approach from "Where Did the Gap Go?" by Tönshoff et al. (2023) starting with default configurations and conducting a linear hyperparameter search. Initial experiments on batch sizes and learning rates showed minimal impact, so we excluded these from further optimization due to computational constraints. On the following hyperparameters we performed linear sweeps to determine the best option. Number of layers, hidden inner dimensions, and pre/post-processing layers. Note that 2 for pre-post_mp corresponds to an MLP with one hidden layer. For each hyperparameter combination in the linear search, we conducted single training run and selected the configuration with the lowest validation LogMSE. The best configuration for each architecture-dataset pair was then trained with three random seeds to obtain the final results reported in Table 2.

Table 4: Hyperparameter search space and selected configurations for all baseline models. We perform model selection over key architectural parameters (layers, hidden dimensions, readout) on LRIM-16 and LRIM-32 datasets. Bold values indicate the best-performing configuration selected based on validation performance for each model architecture. The number of layers was identified as the most impactful parameter and was chosen as a percentage of the system diameter to ensure fair comparison across different graph sizes.

| Dataset | Param | GIN | GatedGCN | GatedGCN-VN$_G$ | GPS-Base | GPS-RWSE | GPS-LapPe |
|---|---|---|---|---|---|---|---|
| lrim_32_1.5 | layers_post_mp | {**2**, 3} | {**2**, 3} | {**2**, 3} | {**2**,3} | {**2**, 3} | {**2**, 3} |
| | layers_pre_mp | {**0**, 2} | {**0**, 2} | {**0**, 2} | {**0**, 2} | {**0**, 2} | {**0**, 2} |
| | dim_inner | {32, 64, **128**} | {32, **64**, 128} | {32, 64, **128**} | {32, **64**, 128} | {**64**, 128} | {32, **64**, 128} |
| | layers | {1, 8, 16, 24, **32**} | {1, 8, 16, **24**, 32} | {1, 8, 16, **24**, 32} | {1, 8, 16, **24**, 32} | {1, 8, 16, **24**, 32} | {1, 8, **16**, 24, 32} |
| lrim_32_0.6 | layers_post_mp | {**2**, 3} | {**2**, 3} | {**2**, 3} | {**2**, 3} | {**2**, 3} | {**2**, 3} |
| | layers_pre_mp | {**0**, 2} | {**0**, 2} | {**0**, 2} | {**0**, 2} | {**0**, 2} | {**0**, 2} |
| | dim_inner | {32, 64, **128**} | {32, 64, **128**} | {32, **64**, 128} | {32, **64**, 128} | {32, 64, **128**} | {32, 64, **128**} |
| | layers | {1, 8, 16, **24**, 32} | {1, 8, 16, 24, **32**} | {1, 8, 16, **24**, 32} | {1, 8, 16, 24, **32**} | {1, 8, 16, **24**, 32} | {1, 8, 16, 24, **32**} |
| lrim_16_1.5 | layers_post_mp | {**2**, 3} | {**2**, 3} | {**2**, 3} | {**2**, 3} | {**2**, 3} | {**2**, 3} |
| | layers_pre_mp | {**0**, 2} | {**0**, 2} | {**0**, 2} | {**0**, 2} | {**0**, 2} | {**0**, 2} |
| | dim_inner | {32, 64, **128**} | {32, 64, **128**} | {32, 64, **128**} | {32, **64**, 128} | {32, **64**, 128} | {32, **64**, 128} |
| | layers | {1, 4, 8, 12, **16**, 20} | {1, 4, 8, 12, 16, **20**} | {1, 4, 8, 12, 16, **20**} | {1, 4, 8, 12, 16, **20**} | {1, 4, 8, 12, **16**, 20} | {1, 4, 8, 12, 16, **20**} |
| lrim_16_0.6 | layers_post_mp | {**2**, 3} | {**2**, 3} | {**2**, 3} | {**2**, 3} | {**2**, 3} | {**2**, 3} |
| | layers_pre_mp | {**0**, 2} | {**0**, 2} | {**0**, 2} | {**0**, 2} | {**0**, 2} | {**0**, 2} |
| | dim_inner | {32, 64, **128**} | {32, **64**, 128} | {32, **64**, 128} | {32, 64, **128**} | {32, 64, **128**} | {32, **64**, 128} |
| | layers | {1, 4, 8, **12**, 16, 20} | {1, 4, 8, 12, **16**, 20} | {1, 4, 8, 12, **16**, 20} | {1, 4, 8, 12, 16, **20**} | {1, 4, 8, 12, **16**, 20} | {1, 4, 8, 12, 16, **20**} |

## E.3  MODEL ARCHITECTURE

We make use of the GraphGPS implementation for all our GPS and MPNN variants. As depicted in Figure 13 the GPS framework mixes local and global message exchange while using appropriate skip-connections, normalizations and feed forward networks similar to the design proposed in (Vaswani et al., 2017). For the RWSE, LapPE or Base variants, the positional encoding part was appropriately modified. The computational cost of computing the LapPE encoding using only $k$ eigenvalues and eigenvectors from Kreuzer et al. (2021) is denoted as $\mathcal{O}(k^2 \cdot E)$, although computing the full spectra would be more expensive. Similarly, the cost of computing the RWSE is

$\mathcal{O}(k \cdot N^2)$ as discussed in Zheng et al. (2025). Whereas both the positional encoding and transformer blocks were skipped for the MPNN implementations where the Local MPNN was either a GIN or GatedGCN.

For the GatedGCN-VN$_G$ baseline, we follow the formulation of Southern et al. (2025), which was built on the same GatedGCN baseline. Similar to the other message-passing baselines, we then encapsulate the GatedGCN-VN$_G$ formulation (as the Local MPNN) inside the GPS layer formulation without any attention blocks. In the following we restate the formulas from Southern et al. (2025) for the general case.

$$\mathbf{h}_{i,\text{loc}}^{(\ell+1)} = \text{up}_\ell^{(\ell)}\left(\mathbf{h}_i^{(\ell)},\ \text{agg}_\ell^{(\ell)}\left(\{\mathbf{h}_j^{(\ell)} : j \in N_i\}\right)\right), \tag{11}$$

$$\mathbf{h}_{\text{vn}}^{(\ell+1)} = \text{up}_{\text{vn}}^{(\ell)}\left(\mathbf{h}_{\text{vn}}^{(\ell)},\ \text{agg}_{\text{vn}}^{(\ell)}\left(\{\mathbf{h}_{j,\text{loc}}^{(\ell+1)} : j \in V\}\right)\right), \tag{12}$$

$$\mathbf{h}_i^{(\ell+1)} = \text{up}^{(\ell)}\left(\mathbf{h}_{i,\text{loc}}^{(\ell+1)},\ \mathbf{h}_{\text{vn}}^{(\ell+1)}\right). \tag{13}$$

And the specific implementation used in combination with GatedGCN.

$$\mathbf{h}_{i,\text{loc}}^{(\ell+1)} = \sigma\left(\Omega^{(\ell)}\mathbf{h}_i^{(\ell)} + \sum_{j \in N_i} \eta^{(\ell)}(\mathbf{h}_i^{(\ell)}, \mathbf{h}_j^{(\ell)}) \odot \mathbf{W}_1^{(\ell)}\mathbf{h}_j^{(\ell)}\right), \tag{6}$$

$$\mathbf{h}_i^{(\ell+1)} = \mathbf{h}_{i,\text{loc}}^{(\ell+1)} + \text{Mean}\left(\{\mathbf{Q}^{(\ell+1)}\mathbf{h}_{j,\text{loc}}^{(\ell+1)}\}_{j \in V}\right), \tag{7}$$

where $\eta^{(\ell)}(\mathbf{h}_i^{(\ell)}, \mathbf{h}_j^{(\ell)}) = \sigma\left(\mathbf{W}_2^{(\ell)}\mathbf{h}_i^{(\ell)} + \mathbf{W}_3^{(\ell)}\mathbf{h}_j^{(\ell)}\right)$

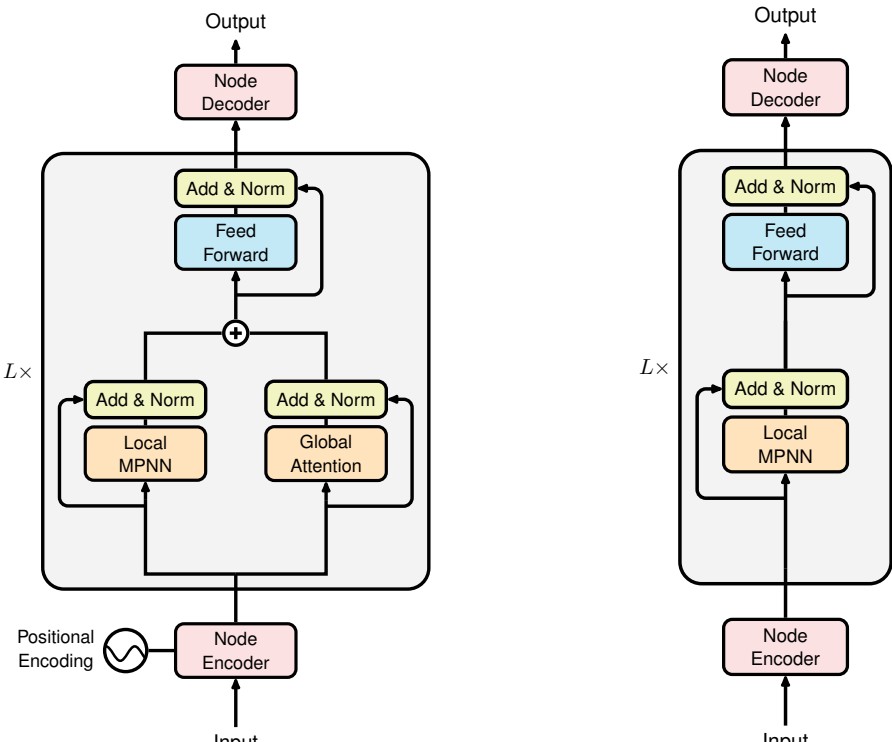

Figure 13: Baseline model architectures used in our evaluation. (Left) Graph transformer (GPS) combines local message-passing with global attention, allowing nodes to directly attend to all other nodes in the graph. (Right) Message-passing neural network (MPNN) rely on local neighborhood aggregation through multiple layers to expand the receptive field. Both architectures use the identical overarching block structure including normalization, skip-connections and Feed Forward Networks stacked $L$ times. Figure design is inspired by the overview of Vaswani et al. (2017).

### E.4 SCALING ABLATION

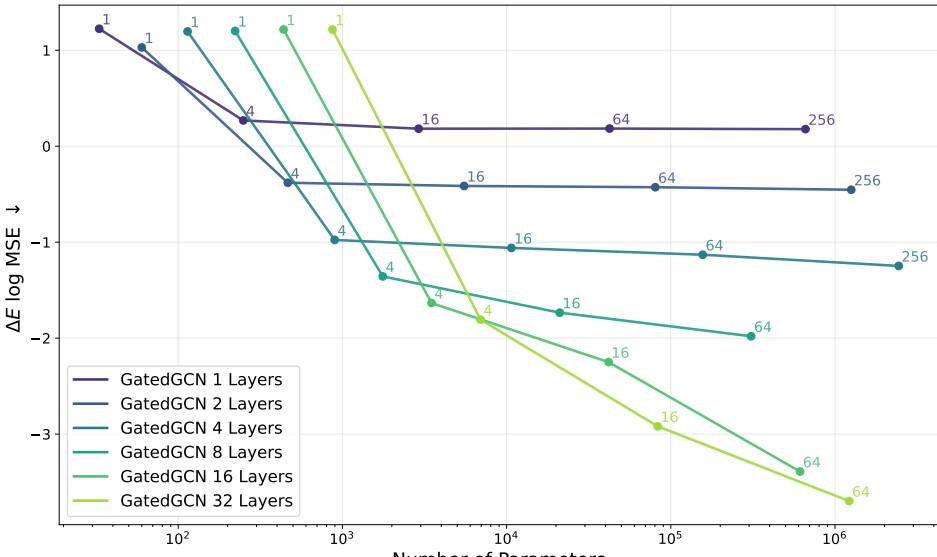

Figure 14: Scaling behavior of GatedGCN models with 1,2,4,8,16 and 32 layers on LRIM-32-hard. The plot shows the logMSE (↓) as a function of the number of parameters across model widths. Each marker is annotated with the corresponding hidden dimension and represents a single seed run configuration.

Table 5: Performance measurement of simple CNN baselines on the LRIM datasets. Each CNN consists of $L$ stacked 2D convolutions of kernel size $K$. They are either using circular padding (Circular-CNN) or zero padding (Grid-CNN). All baselines are trained on the LRIM-16-hard dataset. Additionally, we provide inference results of these models on the LRIM-32-hard and LRIM-64-hard dataset, on which they were not trained, indicated by †. We report results averaged across three seeds.

| Model | Kernel Size | Layers | Width of Receptive Field | LRIM-16-hard ↓ | LRIM-32-hard† ↓ | LRIM-64-hard† ↓ |
|---|---|---|---|---|---|---|
| Circular-CNN | 3 | 1 | 3 | $0.2508 \pm 0.0003$ | $0.3284 \pm 0.0008$ | $0.3506 \pm 0.0011$ |
| | | 2 | 5 | $-0.4437 \pm 0.0002$ | $-0.2521 \pm 0.0007$ | $-0.1946 \pm 0.0006$ |
| | | 3 | 7 | $-0.9829 \pm 0.0020$ | $-0.6570 \pm 0.0004$ | $-0.5536 \pm 0.0002$ |
| | | 5 | 11 | $-1.9050 \pm 0.0091$ | $-1.0236 \pm 0.0106$ | $-0.8133 \pm 0.0078$ |
| | 5 | 1 | 5 | $-0.4444 \pm 0.0008$ | $-0.2530 \pm 0.0006$ | $-0.1953 \pm 0.0008$ |
| | | 2 | 9 | $-1.4562 \pm 0.0016$ | $-0.9009 \pm 0.0076$ | $-0.7384 \pm 0.0063$ |
| | | 3 | 13 | $-2.4117 \pm 0.0016$ | $-1.0488 \pm 0.0062$ | $-0.8184 \pm 0.0041$ |
| | | 5 | 21 | $-2.8905 \pm 0.0376$ | $-1.0552 \pm 0.0103$ | $-0.8169 \pm 0.0065$ |
| | 7 | 1 | 7 | $-0.9904 \pm 0.0009$ | $-0.6584 \pm 0.0011$ | $-0.5537 \pm 0.0009$ |
| | | 2 | 13 | $-2.4080 \pm 0.0103$ | $-1.0484 \pm 0.0047$ | $-0.8185 \pm 0.0033$ |
| | | 3 | 19 | $-2.9319 \pm 0.0638$ | $-1.0652 \pm 0.0073$ | $-0.8227 \pm 0.0053$ |
| | | 5 | 31 | $-2.5400 \pm 0.1229$ | $-1.0501 \pm 0.0090$ | $-0.8130 \pm 0.0059$ |
| Grid-CNN | 3 | 1 | 3 | $0.4088 \pm 0.0002$ | $0.4015 \pm 0.0008$ | $0.3854 \pm 0.0011$ |
| | | 2 | 5 | $0.0209 \pm 0.0001$ | $-0.0207 \pm 0.0007$ | $-0.0728 \pm 0.0010$ |
| | | 3 | 7 | $-0.1443 \pm 0.0002$ | $-0.2309 \pm 0.0014$ | $-0.3254 \pm 0.0018$ |
| | | 5 | 11 | $-0.2498 \pm 0.0011$ | $-0.3779 \pm 0.0055$ | $-0.5019 \pm 0.0124$ |
| | 5 | 1 | 5 | $0.0231 \pm 0.0013$ | $-0.0189 \pm 0.0007$ | $-0.0707 \pm 0.0012$ |
| | | 2 | 9 | $-0.2140 \pm 0.0012$ | $-0.3306 \pm 0.0027$ | $-0.4510 \pm 0.0053$ |
| | | 3 | 13 | $-0.2638 \pm 0.0011$ | $-0.3977 \pm 0.0066$ | $-0.5254 \pm 0.0118$ |
| | | 5 | 21 | $-0.2878 \pm 0.0019$ | $-0.4300 \pm 0.0111$ | $-0.5788 \pm 0.0203$ |
| | 7 | 1 | 7 | $-0.1407 \pm 0.0013$ | $-0.2320 \pm 0.0014$ | $-0.3273 \pm 0.0020$ |
| | | 2 | 13 | $-0.2730 \pm 0.0017$ | $-0.4095 \pm 0.0025$ | $-0.5391 \pm 0.0045$ |
| | | 3 | 19 | $-0.2909 \pm 0.0033$ | $-0.4312 \pm 0.0075$ | $-0.5776 \pm 0.0150$ |
| | | 5 | 31 | $-1.0897 \pm 0.0288$ | $-0.0464 \pm 0.0696$ | $-0.1146 \pm 0.1339$ |

## E.5 COMPUTER VISION ABLATION

Our main intended application for the datasets of the LRIM Graph Benchmark is to test **graph learning methods**. However, because the topology consists of 2D lattices, in principle other methods can be used as well. To show that it is viable in principle, we include a small ablation using proof-of-concept formulations of CNNs and ViTs. We would like to point out that the main aim of our benchmark is to provide a testing bed, or impactful research tool, to precisely study long-range capabilities in graph learning. The aim is to uncover and specify obstacles that need to be studied more closely. Of course, testing methods specifically for grid-structured data is possible as well. Because data generation is known, it is possible to test methods that are more and more aligned (by giving more information, or additional grid bias, or re-using the oracle) in order to obtain better scores. What we believe to be more insightful is to develop and test general graph purpose methods - and then quantify their capabilities on our provided benchmark.

The used CNN and ViT baselines as well as their training setup is very minimalist and is to be taken as proof-of-concept rather than fully optimized baselines. To represent the cyclic grid graph as an image, we duplicate the spins to have 3 channels. All models were trained for 200 epochs using the Adam Optimizer with a learning rate of 0.0003, batch size, and hidden dimension of 64. The CNN architectures consist of a 2D convolution which maps the input to the hidden dimension. Then $L-1$ 2D convs are applied. Finally, an MLP predicts the energy values for each node. The ViT (Dosovitskiy et al., 2021) first splits the grid into square patches, flattens each patch using a linear encoder, then an Encoder Transformer is used for $L$ layers in combination with sinusoidal positional encoding. The tokens are then converted back into patches before an MLP is applied to predict the energy values. We report all results of the CNNs in Table 5 and in Table 6 for the ViT. Additionally, there is a visual depiction of the CNN results according to their size of the receptive field in Figure 15.

Table 6: Performance of a simple Visual Transformer (ViT) on the LRIM datasets. Each ViT slices the circular grid into patches of size $p \times p$, then $L$ encoder transformer layers are applied using 2D sinusoidal positional encoding. We report results averaged across three seeds.

| Model | Patch Size | Layers | Width of Receptive Field | LRIM-16-hard ↓ |
|-------|-----------|--------|--------------------------|----------------|
| ViT | 1 | 1 | 16 | $-1.6473 \pm_{0.0391}$ |
|     |   | 2 | 16 | $-1.7886 \pm_{0.0762}$ |
|     |   | 3 | 16 | $-1.8524 \pm_{0.0982}$ |
|     |   | 5 | 16 | $-2.0629 \pm_{0.1534}$ |
|     | 2 | 1 | 16 | $-0.8017 \pm_{0.0074}$ |
|     |   | 2 | 16 | $-1.7071 \pm_{0.0147}$ |
|     |   | 3 | 16 | $-2.1213 \pm_{0.0381}$ |
|     |   | 5 | 16 | $-2.4185 \pm_{0.0438}$ |
|     | 4 | 1 | 16 | $-0.8205 \pm_{0.0135}$ |
|     |   | 2 | 16 | $-1.4972 \pm_{0.0368}$ |
|     |   | 3 | 16 | $-1.8730 \pm_{0.0378}$ |
|     |   | 5 | 16 | $-2.0481 \pm_{0.0765}$ |

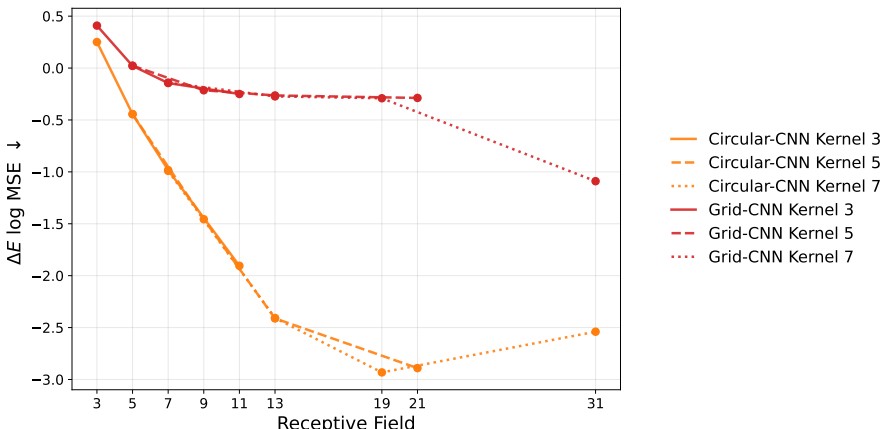

Figure 15: Visual depiction of the obtained logMSE (↓) performance of the CNN baselines reported in Table 5 on the LRIM-16-hard dataset plotted as a function of the width of the receiptive field.

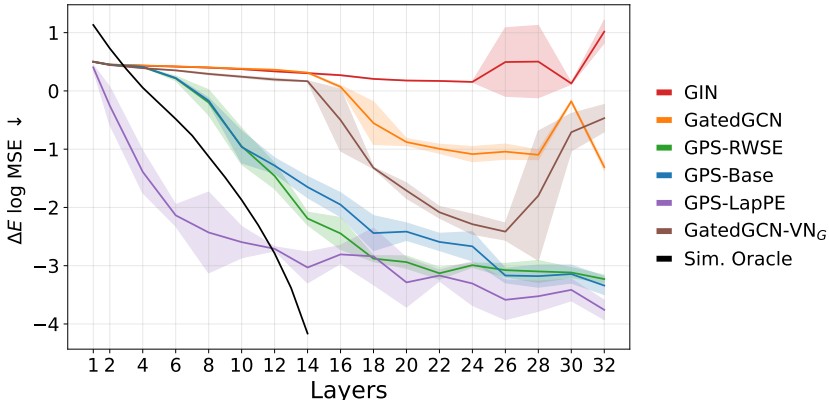

Figure 16: [DIRECTED ABLATION] LogMSE (↓) of the layer ablation study on LRIM-16-hard plotting performance as a function of number of layers. Note, each combination of model and layer is a separate trained model instance. All models consistently improve with increased depth but plateau with a significant gap remaining to the oracle predictor. While we would hope MPNNs to ideally closely follow the oracle as the receptive field expands, they plateau before and training becomes increasingly unstable. Note, that no such trend has to apply to the Graph Transformer as they have access to the entire graph even using a single layer. However, this comes at a signficant computational cost of $\mathcal{O}(LN^2)$ rather than $\mathcal{O}(LE)$.

# F  DIRECTED ABLATION

This section considers a directed variant of our proposed datasets, where edges are directed towards down and right direction, but not the other way around. This drastically alters the receptive fields, especially for the message passing baselines. To distinguish between the two settings we add a [DIRECTED ABLATION] tag to all captions.

Table 7: [DIRECTED ABLATION] Baseline performance on LRIM-16 and LRIM-32 datasets. The number of edges E corresponds to 4N in our datasets. We emphasize the importance of reporting computational complexity alongside performance results, as scalability is a crucial aspect in long-range modeling.

|  | Preprocessing | Computation | LRIM-16-hard ↓ | LRIM-16-easy ↓ | LRIM-32-hard ↓ | LRIM-32-easy ↓ |
|---|---|---|---|---|---|---|
| GIN | - | $\mathcal{O}(L \cdot E)$ | $0.268 \pm 0.008$ | $0.161 \pm 0.011$ | $1.198 \pm 0.026$ | $0.379 \pm 0.004$ |
| GatedGCN | - | $\mathcal{O}(L \cdot E)$ | $-0.871 \pm 0.073$ | $-1.409 \pm 0.052$ | $-0.182 \pm 0.499$ | $0.325 \pm 0.001$ |
| GatedGCN-VN$_G$ | $\mathcal{O}(N)$ | $\mathcal{O}(L \cdot E + L \cdot N)$ | $-1.710 \pm 0.132$ | $-2.266 \pm 0.076$ | $0.669 \pm 0.433$ | $0.238 \pm 0.029$ |
| GPS-Base | - | $\mathcal{O}(L \cdot N^2)$ | $-2.415 \pm 0.149$ | $-2.444 \pm 0.619$ | $-3.098 \pm 0.198$ | $-2.914 \pm 0.291$ |
| GPS-RWSE | $\mathcal{O}(k \cdot N^2)$ | $\mathcal{O}(L \cdot N^2)$ | $-2.938 \pm 0.106$ | $-2.953 \pm 0.367$ | $-3.130 \pm 0.136$ | $-3.547 \pm 0.233$ |
| GPS-LapPE | $\mathcal{O}(k^2 \cdot E)$ | $\mathcal{O}(L \cdot N^2)$ | $-2.758 \pm 0.132$ | $-3.603 \pm 0.438$ | $-3.341 \pm 0.099$ | $-3.448 \pm 0.216$ |

Table 8: [DIRECTED ABLATION] We evaluate how well models trained on LRIM-16-hard transfer to larger systems without additional training. The results show that performance generally degrades as system size increases. GPS variants encounter out-of-memory (OOM) errors on LRIM-256 even for inference-only, demonstrating the importance of considering scalability of long-range methods.

|  | Preprocessing | Computation | LRIM-16-hard ↓ | LRIM-32-hard ↓ | LRIM-64-hard ↓ | LRIM-128-hard ↓ | LRIM-256-hard ↓ |
|---|---|---|---|---|---|---|---|
| GIN | - | $\mathcal{O}(L \cdot E)$ | $0.268 \pm 0.008$ | $0.654 \pm 0.017$ | $0.763 \pm 0.018$ | $0.788 \pm 0.017$ | $0.799 \pm 0.014$ |
| GatedGCN | - | $\mathcal{O}(L \cdot E)$ | $-0.871 \pm 0.073$ | $0.882 \pm 0.012$ | $0.909 \pm 0.013$ | $0.914 \pm 0.012$ | $0.916 \pm 0.012$ |
| GatedGCN-VN$_G$ | $\mathcal{O}(N)$ | $\mathcal{O}(L \cdot E + L \cdot N)$ | $-1.710 \pm 0.132$ | $1.058 \pm 0.111$ | $1.069 \pm 0.075$ | $1.065 \pm 0.085$ | $1.041 \pm 0.147$ |
| GPS-Base | - | $\mathcal{O}(L \cdot N^2)$ | $-2.415 \pm 0.149$ | $-0.950 \pm 0.003$ | $-0.629 \pm 0.029$ | $-0.413 \pm 0.121$ | OOM |
| GPS-RWSE | $\mathcal{O}(k \cdot N^2)$ | $\mathcal{O}(L \cdot N^2)$ | $-2.938 \pm 0.106$ | $-0.966 \pm 0.020$ | $-0.659 \pm 0.021$ | $-0.549 \pm 0.016$ | OOM |
| GPS-LapPE | $\mathcal{O}(k^2 \cdot E)$ | $\mathcal{O}(L \cdot N^2)$ | $-2.758 \pm 0.132$ | $-0.289 \pm 0.625$ | $0.080 \pm 0.688$ | $0.245 \pm 0.732$ | OOM |

## G  RANGE OF THE ORACLE AS A TASK

### G.1  ANALYTICAL EXPRESSION OF LONG-RANGE MEASURE

In this subsection we provide the derivation of Proposition 5.2 and its analytical expression necessary for performing the experiment. We first give an analytical expression of the range measure for a "gradient" of the oracle. Recall that in Bamberger et al. (2025), the range measure of a node-level differentiable function $F$ for a node $u$, denoted by $\rho_u(F)$, is defined as

$$\rho_u(F) = \sum_{v \in V} \left| \frac{\partial F(X)_u}{\partial x_v} \right| d_G(u, v).$$

Here, $x_v$ is the node feature of $v$ and $d_G : V \times V \to \mathbb{R}$ is a distance function on a graph $G = (V, E)$.

Consider a periodic grid graph $G$ of size $L \times L$ to each of whose nodes is assigned an $O(1)$ spin. The diameter $D$ of $G$ is equal to $L$ when $L$ is even and $L - 1$ when $L$ is odd. We consider $\Delta E$ as $F$ and spins as the node feature of nodes. Hereinafter, we will assume that spins $\{s_i\}$ are embedded into the real line $\mathbb{R}$ and consider each spin to be a variable in $\mathbb{R}$. Let also $N_{\leq r}(i)$ be the set of nodes whose shortest path distance from $i$ is shorter than or equal to $r$. Recall that the oracle for a node $i$ with spin $s_i$ is defined as

$$\Delta E_i = s_i \sum_{j \in N_{\leq r}(i)} s_j J_{ij}.$$

Note that we set $J_{ik} = 0$ when $k = i$ as a convention.

**Graph shortest path distance**: We first give in Proposition G.1 the expression of the adapted long-range measures in the case when $d_G : V \times V \to \mathbb{R}$ is the graph shortest path distance of $G$.

**Proposition G.1.** *Let $G$ be a periodic Ising model defined over a grid of size $L \times L$, with $F = \Delta E$. Assume that the binary spins $s_i$ are relaxed to continuous spin values $s_i \in \mathbb{R}$. Then, when $d_G : V \times V \to \mathbb{R}$ is the graph shortest path distance, the range measure $\rho_i(\Delta E)$ of a node-level differentiable function $\Delta E$ for a node $i$ and its normalized measure $\hat{\rho}_i(\Delta E)$ are represented as*

$$\rho_i(\Delta E) = \sum_{0 \leq \ell \leq r} \ell \sum_{k \in N_\ell(i)} J_{ik}, \qquad \hat{\rho}_i(\Delta E) = \frac{\rho_i(\Delta E)}{\sum_{0 \leq \ell \leq r} \sum_{k \in N_\ell(i)} J_{ik}},$$

*in which $N_\ell(i)$ is the set of nodes whose shortest path distance from $i$ is equal to $\ell$.*

*Proof.* The gradient of $\Delta E_i$ with respect to a spin $s_k$ at a node $k$ is

$$\frac{\partial \Delta E_i}{\partial s_k} = \begin{cases} \sum\limits_{j \in N_{\leq r}(i)} s_j J_{ij}, & \text{if } k = i, \\ s_i J_{ij}, & \text{if } k = j \in N_{\leq r}(i), \\ 0, & \text{o.w.} \end{cases} \tag{8}$$

Then, the measure is

$$\rho_i(\Delta E) = \sum_{k \in V} \left| \frac{\partial \Delta E_i}{\partial s_k} \right| d_G(i, k) \tag{9}$$

$$= \sum_{k \in N_{\leq r}(i)} J_{ik} \, d_G(i, k). \tag{10}$$

When $d_G(i, k)$ is the graph shortest path distance, we can group nodes based on the shortest path distance, whose resulting set is $N_\ell(i)$ for each $\ell = d_G(i, k)$. This completes the proof. □

In order to compute $\rho_i(\Delta E)$ and $\hat{\rho}_i(\Delta E)$ efficiently, we will give an analytical expression of $\sum_{k \in N_\ell(i)} J_{ik}$; Let $H$ be an Ising model with size $(L + 1) \times (L + 1)$, that gives rise to $G$ by identifying each node on a boundary segment of $H$ with their respective other side. Without loss of

generality, we consider that the graph $H$ is injectively embedded into the lattice $\mathbb{Z}^2(\subset \mathbb{R}^2)$ in a way that the $L^2$ distance of every pair of neighboring nodes is 1. For $1 \leq \forall \ell \leq D$, let also

$$f^{(\ell)}(m) = \sqrt{m^2 + (\ell - m)^2}, \qquad 0 \leq m \leq \ell.$$

When assuming the coordinate of $i$ is $(0,0)$, for each node $k$ in $H$, there exists at least one $(\ell, m) \in \mathbb{Z}^2$ such that $|\mathbf{r}_i - \mathbf{r}_k| = f^{(\ell)}(m)$, and therefore

$$J_{ik} = \left( f^{(\ell)}(m) \right)^{-(d+\sigma)}.$$

Note that the distance $|\mathbf{r}_i - \mathbf{r}_k|$, which is employed for the definition of $f^{(\ell)}(m)$ and $J_{ik}$, is the $L^2$ Euclidean distance through the mapping from $H$ to $G$.

**Theorem G.2** (Even case). *Assume $L$ is even and $L \geq 4$. Let also $j^{(\ell)}(m) = \left( f^{(\ell)}(m) \right)^{-(d+\sigma)}$. Then,*

$$\sum_{k \in N_\ell(i)} J_{ik} = \begin{cases} 4 \displaystyle\sum_{1 \leq m \leq \ell} j^{(\ell)}(m), & 1 \leq \ell < D/2, \\ 4 \displaystyle\sum_{1 \leq m \leq \ell} j^{(\ell)}(m) - 2\, j^{(\ell)}(0), & \ell = D/2, \\ 4 \displaystyle\sum_{1 \leq m \leq \ell} j^{(\ell)}(m) - 4\, j^{(D/2)}(0) - 8 \displaystyle\sum_{1 \leq m \leq \ell - D/2} j^{(\ell)}(m) \\ \quad + 4 \sum_{m = \ell - D/2} j^{(\ell)}(m), & D/2 < \ell \leq D-1, \\ j^{(\ell)}(0), & \ell = D. \end{cases} \tag{11}$$

**Theorem G.3** (Odd case). *Assume $L$ ($\geq 3$) is odd. Let also $j^{(\ell)}(m) = \left( f^{(\ell)}(m) \right)^{-(d+\sigma)}$. Then,*

$$\sum_{k \in N_\ell(i)} J_{ik} = \begin{cases} 4 \displaystyle\sum_{1 \leq m \leq \ell} j^{(\ell)}(m), & 1 \leq \ell \leq D/2, \\ 4 \displaystyle\sum_{1 \leq m \leq \ell} j^{(\ell)}(m) - 4\, j^{(D/2)}(0) - 8 \displaystyle\sum_{1 \leq m < \ell - D/2} j^{(\ell)}(m), & D/2 < \ell \leq D. \end{cases} \tag{12}$$

*Proof.* Since the system we are dealing with assumes the periodic boundary condition, the gradient is invariant to the translational action of $\mathbb{Z}^2$, i.e., the value remain unchanged with the $\mathbb{Z}^2$-shift action. This implies the gradient is constant across the periodic grid. Therefore, it suffices to prove the argument for one fixed node.

**Even case**: Without loss of generality, we consider the coordinate of the center of $H$ is $(0,0)$. We denote the center by $i$.

For $1 \leq \ell < D/2$, whose visualization is shown in Figure 17 (a), a node $k$ such that $d_G(i, k) = \ell$ lies on one of segments defined by the intersections of 4 lines $x \pm y = \pm \ell$ with arbitrary sign for the plus or minus, in which $(x, y)$ represents the coordinate of $k$ in the lattice $\mathbb{Z}^2$. Since $f^{(\ell)}(m)$ is symmetric (on the segments) with respect to reflection across $x$- and $y$-axis as well as $x \pm y = 0$, we get

$$\sum_{k \in N_\ell(i)} J_{ik} = 4 \sum_{1 \leq m \leq \ell} j^{(\ell)}(m).$$

For $\ell = D/2$, we begin the proof in the non-periodic grid system $H$; see also Figure 17 (b). We first follow the same argument as in the case of $1 \leq \ell < D/2$, and get the same expression as above. When the periodic boundary condition is reimposed on $H$, two nodes located on a horizontal and vertical boundaries of $H$ respectively are identified with nodes on their respective other sides. Therefore, we subtract two contributions, which gives the following contribution for $G$

$$\sum_{k \in N_\ell(i)} J_{ik} = 4 \sum_{1 \leq m \leq \ell} j^{(\ell)}(m) - 2\, j^{(\ell)}(0).$$

For $D/2 < \ell \leq D-1$, we again consider 4 line segments defined by the intersection of $x \pm y = \pm \ell$ with arbitrary sign for the plus or minus. We again start by considering those lines on $H$, but also include contributions of spins lying outside of $H$ which correspond to dotted lines and circles in

Figure 17 (c). We first calculate all the possible contributions on the 4 segments, the total of which is $4 \sum_{1 \le m \le \ell} j^{(\ell)}(m)$. Among the contributions, total contribution from spins on the 4 segments which are located outside of $H$ is

$$4\, j^{(D/2)}(0) + 8 \sum_{1 \le m \le \ell - D/2} j^{(\ell)}(m).$$

The first term corresponds to the contributions of the 4 intersections of the lines and the second term is the sum of contributions corresponding to spins lying outside and boundary of $H$. Therefore, an overall contribution of interior spins of $H$ (excluding boundary spins as well) that are on the segments is

$$4 \sum_{1 \le m \le \ell} j^{(\ell)}(m) - 4\, j^{(D/2)}(0) - 8 \sum_{1 \le m \le \ell - D/2} j^{(\ell)}(m).$$

There remain 8 contributions on the boundary of $H$, each of which is identified with another boundary spin (as in the case of $\ell = D/2$) in $G$. Therefore, we add $4 \sum_{m = \ell - D/2} j^{(\ell)}(m)$ on the boundary, and get the result.

Finally, we have 4 spins whose shortest path distance from $i$ is $D$, which correspond to the corners of $H$. All those points are identified as one node in $G$. Therefore the contribution for $l = D$ is $\left( \dfrac{\sqrt{2}}{n} \right)^{d+\sigma}$.

**Odd case**: We consider that $H$ is embedded in a way that in a manner similar to the case of the even case, but the center of its upper right $L \times L$ subgrid is at $(0,0)$. We denote this subgrid by $\tilde{G}$. Notice that for odd $L$, nodes on the left and the bottom boundaries of $H$ are identified with nodes on the respective other sides whose shortest path distance from $i$ is shorter than that of nodes on the left and bottom edges. Therefore, we can ignore all nodes on the left and bottom boundaries, and nodes from the non-periodic grid $\tilde{G}$ accounts for all contributions for $\rho_i(\Delta E)$.

For $1 \le \ell \le D/2$, we can proceed in a manner similar to that of $1 \le \ell < D/2$ of the even case. Since $\tilde{G}$ is non-periodic, the argument is also valid for $\ell = D/2$. Therefore, the equation

$$\sum_{k \in N_\ell(i)} J_{ik} = 4 \sum_{1 \le m \le \ell} j^{(\ell)}(m)$$

holds for $1 \le \forall \ell \le D/2$.

The derivation of the expression for $D/2 < \ell \le D$ is also straightforward; We apply to $\tilde{G}$ an intermediate argument in the proof of $D/2 < \ell \le D - 1$ in the even case, and we will get

$$\sum_{k \in N_\ell(i)} J_{ik} = 4 \sum_{1 \le m \le \ell} j^{(\ell)}(m) - 4\, j^{(D/2)}(0) - 8 \sum_{1 \le m < \ell - D/2} j^{(\ell)}(m).$$

$\square$

Finally, we give a theoretical guarantee of a threshold of $\sigma$ below which the range measure diverges. This result fairly agrees with the figures reported in reported in Figure 2 and Table 2, and it suggests that in terms of the range measure defined in Proposition G.1, the Ising model could potentially retain infinitely long long-range dependency.

**Proposition G.4.** *Given the same assumption as Proposition G.1 with $r = L$, the range measures $\rho_i(\Delta E)$ and $\hat{\rho}_i(\Delta E)$ diverge as $L \to \infty$ when $\sigma \le 1$, and converge when $\sigma > 1$.*

*Proof.* Without loss of generality, we set $i = (0,0)$. Then,

$$\rho_i(\Delta E) = \sum_{1 \le \ell \le L} \ell \sum_{k \in N_\ell(i)} \frac{1}{|\mathbf{r}_i - \mathbf{r}_k|^{d+\sigma}} \tag{13}$$

$$= \sum_{1 \le \ell \le L} \ell \sum_{k \in N_\ell(i)} \frac{1}{|\mathbf{r}_k|^{d+\sigma}}. \tag{14}$$

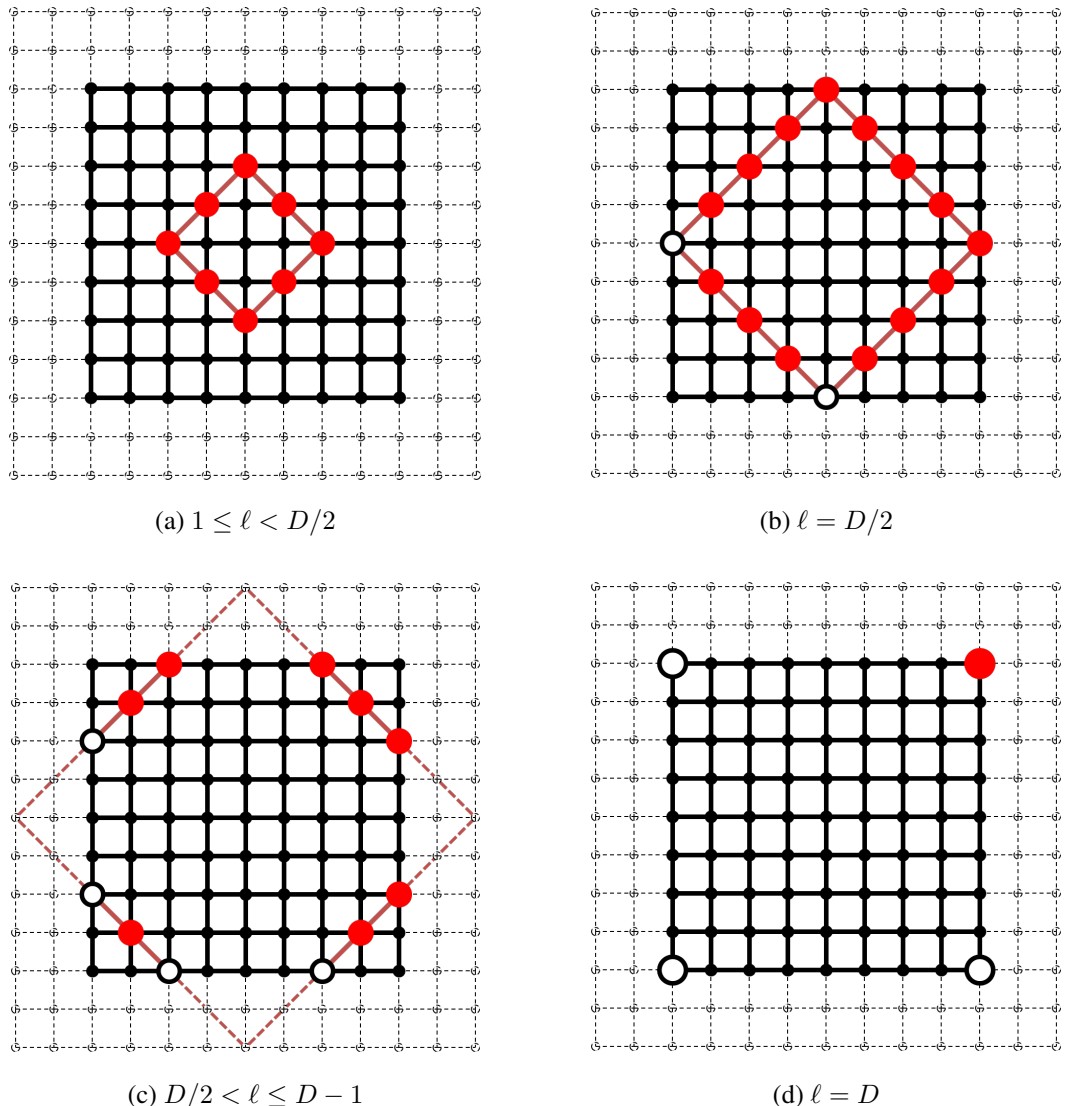

Figure 17: Visualization of contributing spins on $H$ with size $(L+1) \times (L+1)$ for $L = 8 \, (= D)$, depending on graph shortest path distance $d_G(i, k) = \ell$. Dotted circles are complementary virtual spins located outside of $H$ which is drawn by solid lines and circles. Red solid lines represent $x \pm y = \pm \ell$ and red solid circles are spins composing $N_\ell(i)$.

Because for each $\ell$,

$$f^{(\ell)}(m) \le \ell, \quad 0 \le \forall m \le \ell,$$

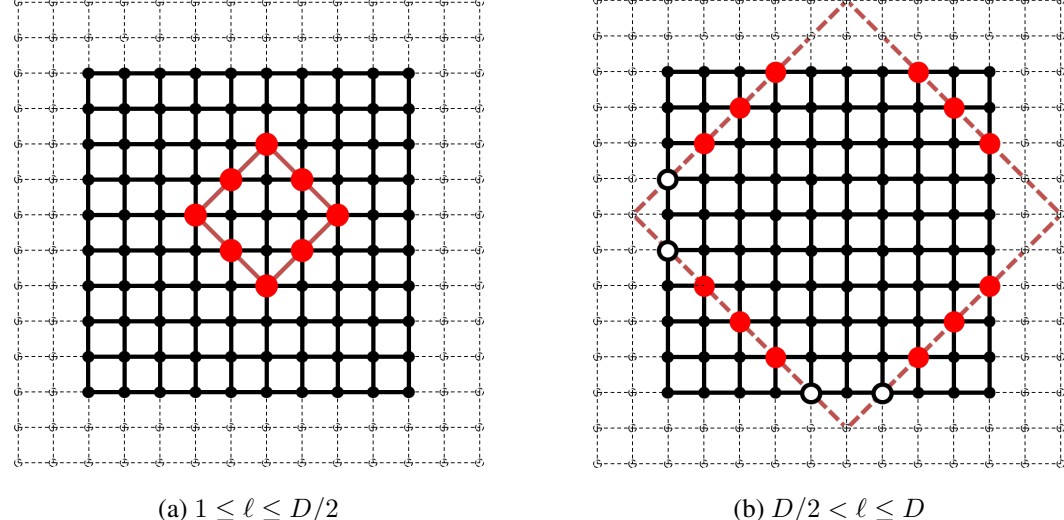

(a) $1 \leq \ell \leq D/2$          (b) $D/2 < \ell \leq D$

Figure 18: Visualization of contributing spins on $H$ with size $(L+1) \times (L+1)$ for $L = 9 \, (= D+1)$, depending on graph shortest path distance $d_G(i,k) = \ell$. Dotted circles are complementary virtual spins located outside of $H$ which is drawn by solid lines and circles. Red solid lines represent $x \pm y = \pm \ell$ and red solid circles are spins composing $N_\ell(i)$.

we get

$$\rho_i(\Delta E) \geq \sum_{1 \leq \ell < D/2} \ell \sum_{k \in N_\ell(i)} \frac{1}{|\mathbf{r}_k|^{d+\sigma}} \tag{15}$$

$$\geq \sum_{1 \leq \ell < D/2} \ell \sum_{k \in N_\ell(i)} \frac{1}{\ell^{d+\sigma}} \tag{16}$$

$$= \sum_{1 \leq \ell < D/2} \ell \cdot 4\ell \frac{1}{\ell^{d+\sigma}} \tag{17}$$

$$= 4 \sum_{1 \leq \ell < D/2} \frac{1}{\ell^\sigma}. \tag{18}$$

The integral test implies that when $\sigma \leq 1$, $\rho_i(\Delta E)$ diverges as $L \to \infty$.

On the other hand, noting that

$$\sqrt{\left(\frac{\ell}{2}\right)^2 + \left(\frac{\ell}{2}\right)^2} \leq \sqrt{m^2 + (\ell - m)^2}, \quad 0 \leq \forall m \leq \ell,$$

we also get the following upper-bound:

$$\rho_i(\Delta E) = \sum_{1 \leq \ell \leq L} \ell \sum_{k \in N_\ell(i)} \frac{1}{|\mathbf{r}_k|^{d+\sigma}} \tag{19}$$

$$\leq \sum_{1 \leq \ell \leq L} \ell \sum_{k \in N_\ell(i)} \left(\frac{\sqrt{2}}{\ell}\right)^{d+\sigma} \tag{20}$$

$$\leq \sum_{1 \leq \ell \leq L} \ell \cdot 4\ell \left(\frac{\sqrt{2}}{\ell}\right)^{d+\sigma} \tag{21}$$

$$= 4\sqrt{2}^{d+\sigma} \sum_{1 \leq \ell \leq L} \frac{1}{\ell^\sigma}. \tag{22}$$

Hence, when $\sigma > 1$, $\rho_i(\Delta E)$ converges as $L$ approaches to $\infty$.

Finally, the normalization term is evaluated from above as follows:

$$\sum_{1 \le \ell \le L} \sum_{k \in N_\ell(i)} J_{i,k} = \sum_{1 \le \ell \le L} \sum_{k \in N_\ell(i)} \frac{1}{|\mathbf{r}_k|^{d+\sigma}} \tag{23}$$

$$\le \sum_{1 \le \ell \le L} \sum_{k \in N_\ell(i)} \left( \frac{\sqrt{2}}{\ell} \right)^{d+\sigma} \tag{24}$$

$$\le \sum_{1 \le \ell \le L} 4\ell \left( \frac{\sqrt{2}}{\ell} \right)^{d+\sigma} \tag{25}$$

$$= 4\sqrt{2}^{d+\sigma} \sum_{1 \le \ell \le L} \frac{1}{\ell^{1+\sigma}}. \tag{26}$$

When $L \to \infty$, the upper bound converges for any $\sigma > 0$, hence so is $\sum_{1 \le \ell \le L} \sum_{k \in N_\ell(i)} J_{i,k}$. $\qquad\square$

## G.2 RANGE MEASURE PLOT VARYING $\sigma$ AND $L$

We now investigate if there is a relationship between the normalized range measure and $\sigma$. Figure 19 displays the range measure values using the analytic expression derived in Section G.1. We use the same configuration as those reported in Table 2 with additional configuration for some parameters: The size of the Ising model ranges in $[16, 32, 64, 128, 256]$, which is exactly same as those in Table 1. We let $\sigma$ takes wider values $[0.3, 0.6, 0.9, 1.2, 1.5, 1.8]$ than the $\sigma$ values in Table 1, to evaluate the impact of the magnitude of $\sigma$ onto the measure. We set $r$ to be same as the diameter of the periodic grid across all the configuration.

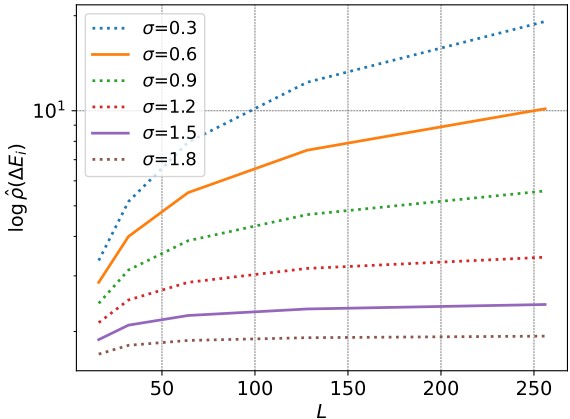

Figure 19: Normalized node range for the graph shortest path distance metric with different sigma. The experiments are run over Ising models with even grid size. $\sigma$ for Ising models for solid lines is identical to that reported in Table 1. The dotted lines correspond to additional $\sigma$ introduced to visualize trend in the long-range metric when varying $\sigma$.

Figure 19 shows a clear upward trend in the measure along the increase in the model size $L$. This result is fairly consistent with a similar trend in the measure evaluated for sparse graphs (Bamberger et al., 2025), where a graph is considered to show longer-range dependency when the graph becomes sparse. We believe the trend in Figure 19 partially accounts for the performance degradation for smaller $\sigma$ and/or larger $L$ as reported in Figure 2 and Table 2.

## H    USAGE OF LARGE LANGUAGE MODELS

We have made use of several Large Language Models (LLMs) during the preparation of this work. ChatGPT and Claude were employed to assist with spellchecking, improving wording, and shortening text for clarity and readability. In addition, ChatGPT, Claude, and Cursor were used for analyzing and explaining code, providing code completions, and generating visualizations to support our implementation and experiments. These tools were applied as auxiliary aids to polish the writing and streamline the development process, while the core research contributions, experimental design, and interpretation of results remain entirely our own.

## I    REPRODUCIBILITY

The source code that we used for our experiments as well as the data for all of the LRIM datasets is available on GitHub[2].

For detailed descriptions of the evaluation or experimental setup we refer to Appendix E.

---

[2]https://github.com/iJorl/lrim_graph_benchmark

