# OpenReview forum: "LRIM: a Physics-Based Benchmark for Provably Evaluating Long-Range Capabilities in Graph Learning"
_ICLR.cc/2026/Conference — ICLR 2026 Poster_

### Official Review · Reviewer_PboE · 2025-10-24

**Soundness:** 3
**Presentation:** 3
**Contribution:** 2
**Rating:** 6
**Confidence:** 2

**Summary:**

The paper constructs a graph regression dataset based on the well-known ising model as a dataset for long-range dependencies.
They show that indeed long-range interactions are needed in order to solve the problem and that deeper networks tend to work a lot better than shallow ones. They also provide an oracle value for $k$-hop GNNs which is a lot better than the trained networks, indicating sufficient complexity to be challenging.

**Strengths:**

The main strength of the suggested dataset is that it addresses one of the key problems in graph machine learning: long-range interactions. Existing datasets are often purely empirical (maybe except for the road networks from Liang et al 2025) instead of being principled.
The dataset satisfies a number of desirable properties such as coming in varying complexities and sizes (including large graphs with 65k nodes each) while leaving a significant performance gap between a simple restricted oracle and existing GNNs.

**Weaknesses:**

Overall, I am not yet convinced that the dataset is really what we are looking for, mostly because the graphs are extremely simple and as far as I understood the task, it is not that much about interactions influencing spin patterns, but rather on aggregating information from far away but in a way that is mostly independent of information that has already been digested.

Concretely, there are a few aspects about the dataset that I consider not too strong:
- it could have been a lot more clear how exactly the LRIM graphs are generated based on the background that has been described before, especially for graph-learning experts that have not worked with the ising model before. Apart from that, the paper is well-written and easy to follow.
- The graphs are extremely simple (just lattices, even simpler than the road networks from Liang et al). Thus the task is really an oversimplified edge case for graph learning.
- There is not that much interaction going on, especially since on a regular lattice all $J_{ij}$ are the same.
- I did not get convinced that the construction really tests interaction and not just global aggregation (see questions).
- The provided lower bound states that there exists a solution thats "very different", but says nothing about the distribution of such solutions. In particular I believe its possible to "cheat" using global statistics.

Concrete (small) things:
- since LRIM uses lattice graphs, positional embeddings should make looking at the edges irrelevant. (e.g. using a PE that is made for images and is able to encode an x and y position)
- The task is really about very exact computations which tends to be not the strongest suite of machine learning models. And at some point floating-point precision will start becoming problematic (probably way before -20 where the trivial accuracy boundary is).
- in the experiments it looked to me that a lot larger MPGNNs would have been possible without exceeding the computational demands of the tested graph transformers. Is there a reason why only "small" models have been tested? How does a 50M GatedGCN model perform?
- 422: Maybe I was misreading the plot, but the numbers of oracle and learned method are not far apart for up to 12 layers (Fig 4). I do not agree with the conclusion made here.
- 475: As soon as we know the distances, the graph itself becomes highly unimportant. So it is only partially about graph learning, i'd say.

**Questions:**

1. is it really "interaction" or rather "aggregation" that is happening in the ising model? Especially when it is about energy prediction as in LRIM?
2. in the monte-carlo simulation that is used to simulate the system, I do not really understand how deterministic this is and how exactly it is used for LRIM.
3. when going for more complex graphs
4. How do you rate the possibility of cheating for a model based on e.g. global statistics and thus outperforming the oracle which has limited information (but uses that information optimally). And in that context, how helpful is the provided lower bound?
5. Do you have an intuition why LapPE did not perform at all? And how it happened to have this odd curve in Fig 4?

---

> ### Author Response · Authors · 2025-11-20
>
> We thank the reviewer for his overall positive assessment of our work.
>
> > it could have been a lot more clear how exactly the LRIM graphs are generated based on the background that has been described before, especially for graph-learning experts that have not worked with the ising model before. Apart from that, the paper is well-written and easy to follow.
>
> > in the monte-carlo simulation that is used to simulate the system, I do not really understand how deterministic this is and how exactly it is used for LRIM.
>
> The simulation itself is inherently stochastic, but due to the use of seeded random number generators the individual runs can be repeated deterministically. To emphasise this interplay, we have added a short sentence outlining the basic idea behind Monte Carlo simulations in the main text. In addition, we have added pseudo code implementation of both standard single spin-flip Monte Carlo and the O(N) cluster algorithm for long-range interacting systems in the appendix B. We hope this clarifies the data generation procedure.
>
> >The task is really about very exact computations which tends to be not the strongest suite of machine learning models. And at some point floating-point precision will start becoming problematic (probably way before -20 where the trivial accuracy boundary is).
>
> The task is mainly about accounting for all relevant (pairwise) interactions. While precision can be a concern as the contributions become smaller, note that even in our empirical results we see a fairly large gap between MPNN and GT, which is unlikely to be caused purely by precision issues. As such we deem it feasible to investigate the current hurdles and develop a better understanding of long-range interactions for graph learning methods. Moreover, the difficulty or size of the datasets could be adjusted to partly mitigate these issues, at least in terms of saturation on smallest instances.
>
> >in the experiments it looked to me that a lot larger MPGNNs would have been possible without exceeding the computational demands of the tested graph transformers. Is there a reason why only "small" models have been tested? How does a 50M GatedGCN model perform?
>
> Our main focus for the evaluation was to test the most common and popular baselines that researchers usually compare against. To do that in a fair and robust way, we carried over a proper  model selection, for which parameter budget was implicitly affected by choosing the hidden dimension and number of layers (see Appendix E.2) that are common in the literature. Therefore, we did not specifically investigate scaling trends for large MPNNs. As a note, we have not observed significant impact from the parameter count alone (during model selection). To provide a small ablation regarding your input, we ran a scaling experiment for GatedGCN in Figure 14 in the Appendix. While scaling does help with performance, especially increasing the number of layers, the training “broke down” (as in did not train well, not unforeseen instabilities during training) if we tried scaling it further.
>
> >422: Maybe I was misreading the plot, but the numbers of oracle and learned method are not far apart for up to 12 layers (Fig 4). I do not agree with the conclusion made here.
>
> Thank you for the input, we will refine our phrasing for the interpretation of Figure 4. The performance of the oracle is what one would get by correctly accounting for all the spins within X hops according to the simulation. As such it is the most important for MPNNs, because they also iteratively expand their neighborhood. The gap between MPNNs and the oracle is quite significant. On the other hand, GT baselines seem to perform better. They see the whole graph immediately thanks to the use of positional encodings and attention mechanisms, and in the case of LapPE can perform better than the oracle for these reasons(up to roughly 12 layers). However the other GTs do not, i.e. at 8 layers the gap between the oracle and Base/RWSE is almost as wide as the gap between them and MPNNs. At the same time the performance trend looks similar to the oracle (with roughly a shift of 2-4 layers) up to ~12-14 layers.
>
> >475: As soon as we know the distances, the graph itself becomes highly unimportant. So it is only partially about graph learning, i'd say.
>
> The necessary distance information can be indeed extracted, which is possible due to the pairwise interactions of the LRIM. These interactions depend on distances and spins (because we know exactly how the data is generated). Our goal is to enable the study of whether graph learning can a) incorporate these long-range dependencies at all and b) how it can do so in a scalable manner. Therefore, using pairwise distances would not satisfy the second requirement. We believe that the most promising way to use our benchmark is to develop and then test general graph purpose methods - and then quantify their capabilities on our provided benchmark.

---

> > ### Author Response · Authors · 2025-11-20
> >
> > >The graphs are extremely simple (just lattices, even simpler than the road networks from Liang et al). Thus the task is really an oversimplified edge case for graph learning.
> > >when going for more complex graphs
> >
> > We agree that the regular topology is one of the current limitations of the provided benchmark. Ultimately, it was a deliberate decision to focus on the well-understood periodic lattice Ising-model, which allows us to generate interesting data at critical temperatures and rigorously analyse and argue about long-range capabilities. As mentioned in our limitations section, we hope to extend this in the future to more general topologies, however, the simulation (particularly, finding the pseudo critical temperature for such systems) and analysis require more investigations and considerations in order to be properly transferred.
> >
> > >There is not that much interaction going on, especially since on a regular lattice all are the same, I did not get convinced that the construction really tests interaction and not just global aggregation (see questions).
> > >is it really "interaction" or rather "aggregation" that is happening in the ising model? Especially when it is about energy prediction as in LRIM?
> >
> > In order to have a node predict its own energy correctly, in principle information across the whole graph has to be aggregated towards that node, so information flow is crucial especially for MPNNs. In particular, it is important to account for the correct pairwise interaction of the energy contribution, for which the pairwise distance and the spin of a node is required. To exemplify, just giving the distribution of spins in the graph (aggregation) is insufficient as this does not capture the distances, and even if it would, the pairwise distance information is important, which differs for different nodes. Of course, if pairwise connections are used, the scalability requirement of our benchmark will not be met.
> >
> > > The provided lower bound states that there exists a solution thats "very different", but says nothing about the distribution of such solutions. In particular I believe its possible to "cheat" using global statistics.
> >
> > We are not sure we fully understand the reviewer’s point. The lower bound result specifically considers methods restricted to local neighborhoods (l 321), and indeed using global statistics such as  Laplacian positional encodings empirically provides an advantage compared to an oracle, at least for the first layers (Figure 4) before the effect vanishes. To the best of our knowledge, there is no method in the Ising-model literature that relies on global statistics to compute energies exactly, but we are happy to verify a specific hypothesis the reviewer has in mind. At the same time, we remark again that the purpose of the benchmark is to attain the lowest error possible while staying within certain computational requirements, which make the usage of the perfect “global statistic”, e.g., pairwise distances as per the known objective function, computationally unfeasible. Please let us know if we misunderstood your comment.
> >
> > >How do you rate the possibility of cheating for a model based on e.g. global statistics and thus outperforming the oracle which has limited information (but uses that information optimally). And in that context, how helpful is the provided lower bound?
> >
> > The oracle is specifically relevant for MPNNs, we should have clarified this better. Our analyses focus on identifying performance gaps in graph learning methods, but it is evident from the GT experiments that using global context can help in comparison to a restricted oracle. However, note that global statistics work up to a certain extent, in particular after X layers performances of the oracle surpass that of methods employing global statistics. We believe that accurate modeling of these interactions cannot rely on global statistics.
> >
> > >Do you have an intuition why LapPE did not perform at all? And how it happened to have this odd curve in Fig 4?
> >
> > We believe there has been a misunderstanding here. The GPS-LapPE is the best performing curve on the plot, we have also adjusted the legend to be more readable now. The LapPE initially outperforms the other baselines, as the (costly) computed LapPE gives more information on distances in the graph, then saturates with an increased number of layers.
> >
> >
> > If there are any further comments or inputs we are happy to continue the discussion. Please have a look at our revised draft, which also includes more ablations and clarifications requested by other reviewers. If you are content with the improvements of our submission we would appreciate it if you can reflect this in your score.

---

> > > ### Comment · Reviewer_PboE · 2025-11-21
> > >
> > > Thanks for the detailed and helpful answers which I really appreciate.
> > >
> > > I believe the main topic is what exactly the dataset should achieve and how it approaches this goal.
> > > Even though the dataset "only" performs global aggregation dependent on distance, it is clear that MPGNNs will struggle here and the dataset is thus a challenging testbed, even though there is no interesting graph structure underlying the dataset. While I would still appreciate a dataset variant that includes more variation in the underlying graph structure, I agree that this comes with additional challenges and that the current dataset already constitutes a big step forward when evaluating long-range aggregation (it still does not really feel like interaction as its really just a pairwise operation which is unaffected by graph structure and any other node).
> > >
> > >
> > > Smaller comments to the remaining points:
> > >
> > > - The scaling behavior of MPGNNs is indeed a topic that is often not looked at in too much detail. In this specific case, I did ask because GTs tend to use a lot of compute and thus consider it interesting to test MPGNNs with comparable compute budgets. Also in my experience GatedGCN tended to scale relatively well with model size (as you also observerd in the new experiments).
> > > - for the experiments: do the various GNNs also make use of PEs?
> > > - Since the gap between GTs and MPGNN+VN architectures persists, the whole part about "cheating" through global statistics (which VNs can aggregate) does not apply as you rightly point out.
> > > - Oracle: thanks for adding the comment that GTs are trivially not bounded by the oracle and thus even empirically able to outperform the oracle.
> > > - This is only a minor comment, but would you mind updating the figures to include "(lower is better)" on the y-axis description? At least for me this makes interpreting graphs at a glance a lot easier.
> > >
> > > One last point, given that there are GTs that include the distance in hops as +1/hops in the softmax attention, do they have a much easier time than other architectures, or is that kind of distance "mismatched"?

---

### Official Review · Reviewer_NHz6 · 2025-10-25

**Soundness:** 4
**Presentation:** 3
**Contribution:** 3
**Rating:** 8
**Confidence:** 4

**Summary:**

The paper introduces LRIM, a physics-based benchmark built on the Ising model that provably depends on long-range interactions, addressing gaps in existing graph-learning benchmarks. Concretely, it proposes a node-regression task to predict per-node energy changes on 2D grid graphs that model different spin configurations of an Ising model. A theoretical analysis shows how the dependence on long-range patterns can be directly controlled in this setting. The provided empirical results report that graphs transformers do significantly outperform local MPNNs in this setting, although at a significantly increased computational cost.

**Strengths:**

I think the suggested task is a valuable addition to the existing set of graph learning benchmarks to study long-range information in a controlled setting. In particular:
1. Obtaining a provably long-range graph learning task from the Ising model is an original idea and addresses the main problem of prior "long-range" benchmarks for which the justification of long-rangedness was purely empirical.
2. The reported results do show a clear separation between local and global architectures.
4. The large graph sizes of up to 65k nodes are challenging for standard graph transformers and seem like a good test bed for developing more efficient long-range architectures.
3. The presentation is clear and key details like hyperparameter budgets are fully provided.

**Weaknesses:**

The restrictions to only using regular 2D grids is a weakness in the context of graph learning, as the main feature of GNNs is their ability to process arbitrary graph structures. I think this is an acceptable weakness for a benchmark that intends to be complementary to "real-world" datasets, but a broader range of graph structures would ultimately be more convincing.

The set of provided baselines also misses MPNNs with virtual nodes [1] as a standard trick to propagate global information in graphs. It would be very interesting how such architectures perform on this dataset, as VNs allow for global information aggregation but lack the pairwise global interactions of transformers that seem to align well with the suggested task.

[1] Gilmer, Justin, et al. "Neural message passing for quantum chemistry." International conference on machine learning. Pmlr, 2017.

**Questions:**

1. What is the numerical range of the regression target $\Delta E_i$? Do these need to be normalized for training?
2. Given that the graphs are currently regular 2D grids, would it be reasonable to use the same task for benchmarking vision models like CNNs or Vision Transformers?

---

> ### Author Response · Authors · 2025-11-20
>
> We thank the reviewer for his overall positive assessment of our work.
>
> We agree that the regular topology is one of the current limitations of the provided benchmark. Ultimately, it was a deliberate decision to focus on the well-understood periodic lattice Ising-model setting, which allows us to generate interesting data at computable pseudo critical temperatures and rigorously analyse and argue about long-range properties.
>
> > The set of provided baselines also misses MPNNs with virtual nodes [1] as a standard trick to propagate global information in graphs. It would be very interesting how such architectures perform on this dataset, as VNs allow for global information aggregation but lack the pairwise global interactions of transformers that seem to align well with the suggested task.
>
> Thank you for your valuable suggestion. We have now included a GatedGCN+VN baseline in all our main comparisons including Table 2, Table 3 and Figure 4. Generally speaking, we observe that adding virtual nodes  improves performance wrt. to the GatedGCN base, although not quite catching up to the transformer. Interestingly, we observe the improvement only for sufficiently large layers - and the training seems to worsen again for very deep formulations.
>
> > What is the numerical range of the regression target? Do these need to be normalized for training?
>
> The predicted values lie roughly in the range of -10 to 10. We have included a visualization of the histogram in the Appendix in Figure 11. We did not experiment with output normalisation during training.
>
> > Given that the graphs are currently regular 2D grids, would it be reasonable to use the same task for benchmarking vision models like CNNs or Vision Transformers?
>
> We would like to clarify that we are not computer vision experts, and thus it is not quite really clear to us if that community would be interested in this specific problem. However, it may be the case for the larger datasets and the transfer learning setup. To show that it is viable in principle, we have included a small ablation using proof-of-concept formulations of CNNs and ViTs. We would like to point out that the main aim of our benchmark is to provide a testing bed, or impactful research tool, to precisely study long-range capabilities in **graph learning**. The aim is to uncover and specify obstacles that need to be studied more closely. Of course testing methods specifically for grid-structured data is possible as well. Because the data generation is known, it is possible to test methods that are more and more aligned (by giving more information, or additional grid bias, or reuse the oracle) in order to obtain better scores, what we believe to be more insightful is to develop and test general graph purpose methods - and then quantify their capabilities on our provided benchmark.
>
> If there are any further comments or inputs we are happy to continue the discussion. Please have a look at our revised draft, which also includes more ablations and clarifications requested by other reviewers.

---

> > ### Comment · Reviewer_NHz6 · 2025-11-26
> >
> > I greatly appreciate the clarifications and updates to the manuscript.
> >
> > I think this is a good paper and a valuable contribution to the field of graph learning. I will keep my score as is (accept).

---

### Official Review · Reviewer_6zXs · 2025-11-01

**Soundness:** 3
**Presentation:** 3
**Contribution:** 2
**Rating:** 6
**Confidence:** 3

**Summary:**

This paper introduces a physics based dataset to measure long range modeling capabilities in graph neural networks, named as Long-Range Ising Model (LRIM) Graph Benchmark. The benchmark utilizes the Ising model with power law interactions, where the target task provably depends on long-range dependencies. The paper provides 10 datasets ranging from 256 to 65k nodes with controllable difficulty through tunable parameter that controls the interaction strength between nodes inversely. Analysis shows that local information is insufficient, theoretical study is given on long rangeness measures, and empirical evaluations demonstrate that both message passing architectures and graph transformers perform lower. The entire dataset is synthetically generated and the graphs are 4-regular and 2D grid like.

**Strengths:**

- For the datasets, the use of the Ising model provides a physics based foundation where long range dependencies are mathematically guaranteed and controllable. This is unlike some prior long range benchmarks in graphs such as superpixels where the long range is not mathematically guaranteed.
- Compared to previous benchmarks which demonstrate long rangeness of tasks using performance of different model classes, this work has elaborate analysis of the proposed dataset with oracle predictor, theoretical lower bounds and long rangeness metric.
- The task difficulty can be tuned and is also demonstrated with examples in Figure 3. In addition, there are clear performance gaps between message passing networks and full-neighborhood graph transformers, as in Tables 2,3.
- The proposed collection of datasets with sizes and difficulty can be used for developing long range graph networks, alongside other recent works/datasets which study this topic.

**Weaknesses:**

- As acknowledged by the paper, the benchmark is limited to regular lattice structures. This is significant since real world graphs rarely have such regular topology and and message passing GNNs may not be the best architecture;
the grid structure may favor certain architectural choices.
- In addition, methods designed specifically for grid-like data are excluded. however, including them could inform on the necessity of graph-specific networks perform in such settings.
- A major limitation is the real-world applicability of the datasets which the paper acknowledges.

**Questions:**

One observation, GPS shows OOM on LRIM-256 in Table 3. Is there a possibility to include a more approximate alternative for GPS, for instance, specifically to inform the missing scores here?

---

> ### Author Response · Authors · 2025-11-20
>
> We thank the reviewer for his overall positive assessment of our work.
>
> >As acknowledged by the paper, the benchmark is limited to regular lattice structures. This is significant since real world graphs rarely have such regular topology and and message passing GNNs may not be the best architecture; the grid structure may favor certain architectural choices.
>
> As we have outlined in the limitations section of our submission, the real-world applicability as well as the regular periodic graph topologies are valid concerns. We would like to emphasize that these are byproducts of our deliberate focus on creating a provable benchmark for assessing long-range capabilities in graph learning. The well-understood function used to generate the LRIM data limits real-world applicability, but enables us to further analyse and be precise about the impact and limitations of long-range effects. Similarly, the behaviour of the LRIM model on 2D lattices is well studied and as such a reasonable choice for a topology, unfortunately, application to more general graph topologies requires significant adjustment of the simulation to guarantee appropriate data (as the pseudo critical temperature would need to be computed separately for each topology) and ensure that the dynamic topologies can still provide the wanted long-range analysis akin to our section 5.
>
> > In addition, methods designed specifically for grid-like data are excluded. however, including them could inform on the necessity of graph-specific networks perform in such settings.
>
> We have included an ablation of simple CNNs and ViT methods usually used for computer vision in Appendix E.
> We would like to point out that the main aim of our benchmark is to provide a testing bed, or research tool, to precisely study long-range capabilities in graph learning. The aim is to uncover and specify obstacles that need to be studied more closely. Of course testing methods specifically for grid-structured data is possible as well, but this is orthogonal to our purpose. Because the data generation is known, it is possible in principle to test methods that are more and more aligned (by giving more information, or additional grid bias, or reuse the oracle) with the task, but this defies the purpose of developing and testing more effective and general purpose graph methods.
>
> > One observation, GPS shows OOM on LRIM-256 in Table 3. Is there a possibility to include a more approximate alternative for GPS, for instance, specifically to inform the missing scores here?
>
> Regarding GPS OOM: We double checked the runs on the largest sizes, which we ran on an A100 80GB GPU on LRIM-128 with batch size 4 using roughly 50% of the available VRAM. Using these figures we estimate that a naive run for the GPS models on LRIM-256 (x4 larger instances, resulting in x16 increased memory requirements because of the attention) with batch size 1 would roughly require ~160GB of VRAM. Reducing memory requirement using quantization is unlikely to be of much help, as it is not the weight but the quadratic attention that is the bottleneck. We are currently investigating if it is possible to mitigate the naive realization of attention and make use of more efficient (akin FlashAttention) implementation to reduce memory requirements. In the paper, we have clarified that the culprit for OOM is the naive attention mechanism and will try our best to adapt the code in order to provide further results during the rest of the discussion.
>
> Please let us know if there are any further comments or inputs, we are happy to continue the discussion. Our revised draft now also includes more ablations and clarifications requested by other reviewers. If you are content with the improvements of our submission we would appreciate it if you can reflect this in your score.

---

### Author Response · Authors · 2025-11-28

Dear Reviewers,

We greatly appreciate the constructive feedback received and your help in improving the manuscript by suggesting new analyses that further confirm the validity and the usefulness of the benchmark.

Overall, we have updated the paper by adding, in the appendix, new scaling results that explore the interplay between number of message passing layers and model size. We have also implemented and evaluated an additional message-passing architecture using virtual nodes. As suggested, we have included proof-of-concept experiments with non-MPNN architectures such as CNNs and Transformer architectures. Finally, we have revised the text and figures to improve readability as the reviewers suggested.

We have also resolved an issue related to the empirical analysis, about the use of directed instead of undirected graphs, which affected the quantitative scores of the baselines meant for future comparisons. The design and principled theoretical analysis of our datasets are not affected and our conclusions regarding the results remain the same, but we wanted to let you know that we updated the manuscript accordingly.

The LRIM Graph Benchmark is designed to provide a provable and physics-grounded benchmark for evaluating long-range range capabilities across task difficulty and multiple graph scales, and we thank the reviewers again for their help and support of this work.

Best regards,
The authors

---

### Meta-Review · Area_Chair_W7eU · 2026-01-04

**Summary:**

This submission introduces LRIM, a physics-grounded benchmark for evaluating long-range dependency modeling in graph learning, built from the long-range Ising model. The benchmark formulates a node-level regression task that predicts per-node energy changes on 2D periodic grid graphs under controllable long-range interactions. The benchmark provides 10 datasets spanning 5 graph sizes and 2 difficulty levels per size controlled by the interaction parameter, with performance measured via log-MSE.

Across reviewers, there is broad agreement that the paper’s core strength is the principled construction of a task whose dependence on long-range information is provable and tunable, and that the benchmark meaningfully exposes gaps between local MPNNs and global architectures. Reviewers also appreciated the scalability aspect, which makes the benchmark a useful stress test for efficient long-range graph methods.

The main concerns center on ecological validity and structural diversity: LRIM currently uses highly regular lattice graphs, raising questions about how well the benchmark reflects challenges in real-world graph learning, and whether the task is closer to "global aggregation by distance" than rich graph-structured interaction. In rebuttal, the authors addressed several actionable issues by (i) adding new scaling/ablation results, (ii) implementing a virtual-node MPNN baseline, (iii) adding proof-of-concept CNN/ViT baselines, (iv) improving readability, and (v) correcting an analysis issue related to directed vs. undirected graphs that affected baseline scores while leaving the benchmark design and theoretical analysis unchanged. One reviewer explicitly confirmed that the clarifications and updates were sufficient to retain an accept score.

**Reviewer Concerns:**

**Reviewer 6zXs**

Addressed: The rebuttal clarifies baseline limitations and provides additional analyses (including more scalable alternatives where possible), and explains why some transformer-style baselines can be memory-limited on larger instances.

Outstanding: The benchmark’s restriction to regular lattice structure and limited real-world applicability remain key limitations; reviewer also noted that inclusion of grid-specialized methods could further contextualize performance.

**Reviewer NHz6**

Addressed: Missing baseline with virtual nodes was directly addressed via a GatedGCN+VN baseline added to main comparisons; authors also clarified practical details such as the regression target scale and added supporting appendix analysis.

Outstanding: The reviewer’s primary methodological concern remains: broader graph-structure diversity beyond 2D grids would make the benchmark more convincing for general graph learning.

**Reviewer PboE**

Addressed: The rebuttal discussion helped sharpen the paper’s positioning: as a challenging testbed for long-range aggregation where MPNNs struggle, while clarifying issues around "cheating" via global statistics and the interpretation of oracle comparisons.

Outstanding: The reviewer still questions whether the task captures the kind of graph-dependent interactions the community ultimately wants, and would prefer variants with richer structural variability.

**Reviewer Scores:**

Reviewer 6zXs: 6 (marginally above threshold; no score change indicated).

Reviewer NHz6: 8 (accept, good paper; explicitly kept score after rebuttal).

Reviewer PboE: 6 (marginally above threshold; no score change indicated).

---

### Decision · Program_Chairs · 2026-01-26

Accept (Poster)